# Diamond Maps: Efficient Reward Alignment via Stochastic Flow Maps

**Peter Holderrieth** [1][*]  **Douglas Chen** [2][*]  **Luca Eyring** [3][*]  **Ishin Shah** [2]  **Giri Anantharaman** [2]  **Yutong He** [2]
**Zeynep Akata** [3]  **Tommi Jaakkola** [†][1]  **Nicholas Matthew Boffi** [†][2]  **Max Simchowitz** [†][2]

## Abstract

Flow and diffusion models produce high-quality samples, but adapting them to user preferences or constraints post-training remains costly and brittle, a challenge commonly called reward alignment. We argue that efficient reward alignment should be a property of the generative model itself, not an afterthought, and redesign the model for adaptability. We propose *Diamond Maps*, stochastic flow map models that enable efficient and accurate alignment to arbitrary rewards at inference time. Diamond Maps amortize many simulation steps into a single-step sampler, like flow maps, while preserving the stochasticity required for optimal reward alignment. This design makes search, Sequential Monte Carlo, and guidance scalable by enabling efficient and consistent estimation of the value function. Our experiments show that Diamond Maps can be learned efficiently via distillation from GLASS Flows, achieve stronger reward alignment performance, and scale better than existing methods. Our results point toward a practical route to generative models that can be rapidly adapted to arbitrary preferences and constraints at inference time.

## 1. Introduction

Diffusion models (Song et al., 2020b; Ho et al., 2020) and flow matching (Lipman et al., 2022; Albergo et al., 2023; Liu et al., 2022) are the state-of-the-art generative models for image, video, audio, and molecules. These models are trained to return samples from a *data distribution* $p_{\text{data}}(z)$ over $z \in \mathbb{R}^d$. In many applications, however, we want more than faithful samples: we want to steer them to better satisfy a task objective, e.g. improved text–image alignment. Given a reward function $r(z)$, we seek samples that achieve

high reward and are likely under the data distribution $p_{\text{data}}$. This is commonly referred to as **reward alignment** (Uehara et al., 2024; 2025).

Existing reward alignment methods fall into two broad categories: (1) *Reward fine-tuning* adds an extra training stage to specialize a generator to $r(z)$ (Fan et al., 2023; Wallace et al., 2024; Uehara et al., 2024; Domingo-Enrich et al., 2024). (2) *Inference-time reward alignment* or *guidance* modifies the sampling procedure while leaving the pretrained model unchanged (Uehara et al., 2025; Chung et al., 2022; He et al., 2023; Skreta et al., 2025; Singhal et al., 2025). Both approaches have drawbacks: reward fine-tuning is complex and costly, and must be repeated for each new reward, limiting flexibility. Guidance avoids retraining, but typically relies on approximations that are inaccurate and often increases sampling cost substantially. It remains an open question how to combine the merits of both approaches.

In this work, we choose a different approach, proposing instead to re-design models towards adaptability to arbitrary rewards. The central question we address is:

*How can we build a generative model that enables fast and accurate reward alignment at inference time?*

It is known that exact guidance relies on estimation of the **value function** associated with a reward (Uehara et al., 2025). Intuitively, the value function measures how much reward a given state will return in the future on average. However, estimation of the value function is thought to be intractable with flow and diffusion models, as it requires many simulation steps with an SDE or ODE model (Song et al., 2020b; Holderrieth et al., 2025). We challenge this assumption in this work.

To this end, we propose **Diamond Maps**. Diamond Maps leverages accelerated sampling methods (Song et al., 2023c; Frans et al., 2024), namely flow maps (Boffi et al., 2025a;b; Geng et al., 2025), which allow us to amortize many inference-steps of flow and diffusion models into a single neural network evaluation. In this work, we investigate how to **leverage flow maps to accurately and efficiently estimate the value function** and thereby enable exact guidance.

Our key technical observation is that this can be achieved via *stochastic* **flow maps**. While standard flow maps are

---

[*]Equal contribution † Equal advising [1]MIT CSAIL [2]Carnegie Mellon University [3]TU Munich, Helmholtz Munich, MCML. Correspondence to: Peter Holderrieth <phold@mit.edu>.

*Proceedings of the 43rd International Conference on Machine Learning*, Seoul, South Korea. PMLR 306, 2026. Copyright 2026 by the author(s).

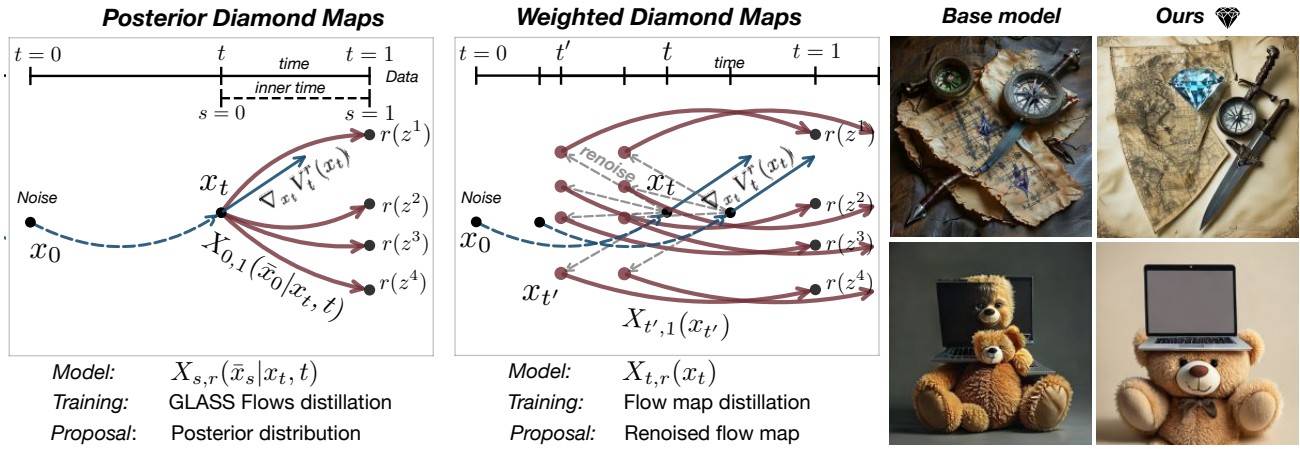

*Figure 1.* **Overview of Diamond Maps.** Diamond Maps are stochastic flow maps that allow to perform one-step "look-aheads" of a flow trajectory (blue) to potential endpoints at time 1 to evaluate a reward $r$. This allows for efficient exploration, search, and guidance. We propose 2 Diamond Map Designs. *Posterior Diamond Maps* (left) distill GLASS Flows into a flow map $X_{s,s'}(\bar{x}|x_t, t)$ designed to sample exact samples from the posterior. *Weighted Diamond Maps* (middle) allow to use standard flow maps by making them stochastic via a simple renoising procedure. Right: Improved image alignment with Diamond Maps. Prompts: "A diamond, a folded treasure map, a compass, and a dagger". "A laptop on top of a teddy bear".

deterministic, recently **GLASS Flows** (Holderrieth et al., 2025) demonstrated that one can easily obtain stochastic transitions from a pre-trained flow model. This opens up the possibility of *flow map distillation*, which we leverage in this work. In addition, we also present a method to make standard flow maps stochastic at inference time, thereby enabling efficient value function estimation.

We make the following contributions (see Figure 1):

1. We introduce **Diamond Maps**, stochastic flow maps for scalable reward alignment. We propose two designs.

2. First, we present **Posterior Diamond Maps**, one-step posterior samplers distilled from GLASS Flows. These offer a simple and effective value function estimation.

3. Second, we present **Weighted Diamond Maps**, an algorithm to turn standard flow maps into consistent value function estimators. This method allows to use off-the-shelf distilled models.

4. We show that Posterior Diamond Maps - despite only trained to sample from posteriors - are also **one-step samplers of the time-reversal SDE** (Song et al., 2020b) enabling scalable proposals for search/SMC.

5. We empirically demonstrate that Diamond Maps provide fast and high-quality reward alignment.

## 2. Background

### 2.1. Flow Matching

We follow the flow matching framework (Lipman et al., 2022; Albergo et al., 2023; Liu et al., 2022), though we

remark that everything applies similarly to score-based diffusion models (Song & Ermon, 2019; Song et al., 2021). We denote data points with vectors $z \in \mathbb{R}^d$ and the *data distribution* with $p_{\text{data}}$. Here, $t = 0$ corresponds to noise ($\mathcal{N}(0, I_d)$) and $t = 1$ corresponds to data ($p_{\text{data}}$). To noise data $z \in \mathbb{R}^d$, we use a *Gaussian conditional probability path* $p_t(x_t|z) = \mathcal{N}(x_t; \alpha_t z, \sigma_t^2 I_d)$:

$$x_t = \alpha_t z + \sigma_t \epsilon, \quad \epsilon \sim \mathcal{N}(0, I_d) \quad (1)$$

where $\alpha_t, \sigma_t \geq 0$ are *schedulers* with $\alpha_0 = \sigma_1 = 0$ and $\alpha_1 = \sigma_0 = 1$ and $\alpha_t$ (resp. $\sigma_t$) strictly monotonically increasing (resp. decreasing). Making $z \sim p_{\text{data}}$ random, this induces a corresponding *marginal probability path* $p_t(x_t) = \mathbb{E}_{z \sim p_{\text{data}}}[p_t(x_t|z)]$ which interpolates Gaussian noise $p_0 = \mathcal{N}(0, I_d)$ and data $p_1 = p_{\text{data}}$. The **flow matching posterior** is defined as

$$p_{1|t}(z|x_t) = \frac{p_t(x_t|z)p_{\text{data}}(z)}{p_t(x_t)} \quad (2)$$

and describes the distribution of clean data $z$ given noisy points $x_t$. Flow matching models learn the *marginal vector field* $u_t(x_t) = \mathbb{E}_{z \sim p_{1|t}(\cdot|x)}[u_t(x|z)]$ where $u_t(x_t|z)$ is the conditional vector field (see Section A.1 for formula). Simulating an ODE with the marginal vector field from initial Gaussian noise leads to a trajectory whose marginals are $p_t$:

$$x_0 \sim p_0, \quad \frac{\mathrm{d}}{\mathrm{d}t}x_t = u_t(x_t) \quad \Rightarrow \quad x_t \sim p_t \quad (3)$$

In particular, $X_1 \sim p_{\text{data}}$ returns a sample from the desired distribution. In this work, we also use the alternative parameterization of the vector field $u_t(x)$ given by the *denoiser* $D_t$, i.e. the expectation of the posterior:

$$D_t(x) = \int z p_{1|t}(z|x)\mathrm{d}z = \frac{\sigma_t u_t(x_t) - \dot{\sigma}_t x_t}{\dot{\alpha}_t \sigma_t - \alpha_t \dot{\sigma}_t} \quad (4)$$

## 2.2. Flow Maps

Flow matching requires the step-wise simulation of an ODE, a computationally expensive procedure. To address this, flow maps (Boffi et al., 2025a;b; Geng et al., 2025) can be used that aim to amortize many simulation steps into a single neural network evaluation. Specifically, the **flow map** $X_{t,t'}(x_t)$ $(x_t \in \mathbb{R}^d, 0 \le t \le t' \le 1)$ is defined as the function that maps from time $t$ to time $t'$ along ODE trajectories, i.e. if $(x_t)_{0 \le t \le 1}$ is an ODE trajectory in (3), then $X_{t,t'}(x_t) = x_{t'}$. Hence, learning $X_{t,t'}$ directly can significantly save inference-time compute. One can then sample from a flow map in a single step by drawing $x_0 \sim \mathcal{N}(0, I_d)$ and setting $x_1 = X_{0,1}(x_0)$, which by construction satisfies $x_1 \sim p_{\text{data}}$.

A diversity of methods have been proposed to learn flow maps either by distillation from existing flow and diffusion models or from scratch via *self-distillation* (Song et al., 2023c; Boffi et al., 2025a;b; Geng et al., 2025). We note that **our method can be used for any distillation method**. As an example, we use the *Lagrangian* loss (Boffi et al., 2025a;b; Zhou et al., 2025; Tong et al., 2025) given by:

$$\mathcal{L}_{\text{Lag}}(\theta) = \mathbb{E}\left[\|\partial_{t'} X_{t,t'}^\theta(x_t) - \text{sg}(u_{t'}(X_{t,t'}(x_t)))\|^2\right] \quad (5)$$

where the expectation is over $z \sim p_{\text{data}}, x_t \sim p_t(\cdot|z)$ and "sg" denotes the stop-gradient operator.

# 3. Value function estimation via Stochastic Flow Maps

In many applications, the distribution $p_{\text{data}}$ that a flow matching model aims to sample from serves only as a *prior* distribution. Often, we are given an objective function $r : \mathbb{R}^d \to \mathbb{R}$ called the **reward function** that we aim to maximize. This could be human preferences, constraints, inpainting, and many other tasks. In this setting, the pre-trained generative model allows us to "regularize" our search space. While some works keep the goal vague as "maximizing $r$ while staying on the data manifold", it is commonly formalized as sampling from the **reward-tilted distribution**

$$p^r(z) = \frac{1}{Z} p_{\text{data}}(z) \exp(r(z)). \quad (6)$$

A central object for inference-time reward alignment is the **value function** (Uehara et al., 2024; 2025) given by

$$V_t(x_t) = \log \mathbb{E}_{z \sim p_{1|t}(\cdot|x_t)}[\exp(r(z))]. \quad (7)$$

Intuitively, the value function allows to *evaluate* states $x_t$ based on the expected reward in the future. The form $\log \mathbb{E}[\exp(\cdot)]$, is not arbitrary but is inherent to the construction of flow and diffusion models as denoising a Gaussian kernel $p_t(x|z)$: the value function satisfies (see Section A.2)

$$\exp(V_t(x_t)) \propto \frac{p_t^r(x_t)}{p_t(x_t)} \quad (8)$$

where $p_t^r(x_t) = \mathbb{E}_{z \sim p^r}[p_t(x_t|z)]$ is the noised reward-tilted distribution. This expression measures how much more likely a *noisy* state $x_t$ is under $p^r$ compared to $p_{\text{data}}$ after noising. The value function $V_t(x_t)$ is the core object that needs to be estimated for most reward alignment methods that target to sample from $p^r$, as we illustrate next.

**Guidance.** A natural approach to sample from $p^r(z)$ is to estimate the corresponding flow matching model $u_t^r(x)$[1]. This is called **(gradient) guidance**. One can show that $u_t^r$ has the following formula (see Section A.2 for a proof):

$$u_t^r(x) = u_t(x) + b_t \nabla_{x_t} V_t(x_t), \quad (9)$$

where $u_t(x)$ is the marginal vector field of the original distribution (non-tilted), and where $b_t = \sigma_t^2 \frac{\dot{\alpha}_t}{\alpha_t} - \dot{\sigma}_t \sigma_t$. As we know $u_t(x)$, accurate guidance is equivalent to accurate estimation of the gradient $\nabla_{x_t} V_t(x_t)$ of the value function.

**Sequential Monte Carlo (SMC) and search** methods evolve a population of particles $x_t^1, \cdots, x_t^K$ from $t = 0$ to $t = 1$ and assign each of them a weight $w_t^1, \cdots, w_t^K$. If $x_t^i \sim p_t$, then by equation (8) setting $w_t^i = \exp(V_t(x_t^i))$ converts a population of particles from $p_t$ to a population of particles from $p_t^r$. Therefore, accurate and efficient estimation of the value function $V_t(x_t)$ enables optimal filtering of particles, facilitating search over the support of the prior.

**Denoiser approximation.** For standard flow and diffusion models the value function is unfortunately *intractable*. The reason for this is that sampling from the posterior $p_{1|t}$ requires simulating a trajectory of an SDE (Song et al., 2020b; Ho et al., 2020) or ODE (Holderrieth et al., 2025), which is too expensive. For this reason, most methods revert to a simple approximation of the posterior $p_{1|t}$ via the denoiser

$$V_t(x_t) \approx r(D_t(x_t)) \quad (10)$$

However, it is well-known that this approximation is highly biased, e.g. leading to the so-called *Jensen gap* in the context of linear inverse problems (Chung et al., 2022). This makes many inference-time adaptation methods inaccurate, and limits applicability to simple rewards.

**Our approach.** In this work, we revisit the value function estimation problem. Instead of reverting to approximations, our goal is to find both (1) consistent (i.e. statistically accurate) and (2) efficient estimators using flow maps. Current sampling with flow maps is *deterministic*, the next point $x_{t'} = X_{t,t'}(x_t)$ is completely determined by $x_t$. However, to truly estimate the value function, we need to take into account *all* possible futures. In other words, we **require stochasticity for exploration**. This motivates our approach:

---

[1]replace $p_{\text{data}}$ with $p^r$ in equ. (2) and the definition of the marginal vector field

**Goal 1.** Estimate the value function $V_t(x_t)$ and its gradient $\nabla_{x_t} V_t(x_t)$ efficiently at inference-time for arbitrary rewards by constructing stochastic flow maps.

Further, we also construct stochastic flow maps for another reason: SMC and search methods construct search trees by stochastic transitions allowing to explore different branches. We want to construct search trees more efficiently:

**Goal 2.** Construct stochastic flow maps that enable fast sampling of stochastic Markov transitions for efficient search and SMC.

Next, we present 2 stochastic flow map designs: (1) *Posterior Diamond Maps* and (2) *Weighted Diamond Maps*.

## 4. Posterior Diamond Maps

In this section, we present *Posterior Diamond Maps*, a stochastic flow map model designed to be efficiently adaptable to a wide range of rewards at inference time. A **Posterior Diamond Map** model $X_{s,s'}(\bar{x}_s|x_t, t)$ is a flow map designed to sample from the posterior distribution $p_{1|t}(\cdot|x_t)$ given $x_t$ ((2)). Note that $x_t$ is fixed for this flow map. Instead, the times $0 \leq s \leq s' \leq 1$ refer to an "inner time axis" that describes the flow evolving from initial Gaussian noise to a sample from $p_{1|t}(\cdot|x_t)$. We sample from this model via

$$\bar{x}_0 \sim \mathcal{N}(0, I_d) \quad \Rightarrow \quad X_{0,1}(\bar{x}_0|x_t, t) \sim p_{1|t}(\cdot|x_t) \quad (11)$$

This enables efficient value function estimation:

**Proposition 4.1.** *For $\bar{x}_0^k \sim \mathcal{N}(0, I_d)$ and $z^k = X_{0,1}(\bar{x}_0^k|x_t, t)$, the following are consistent estimators of the value function and its gradient:*

$$V_t(x_t) \approx \log \frac{1}{N} \sum_{k=1}^{K} \exp(r(z^k)) \quad (12)$$

$$\nabla_{x_t} V_t(x_t) \approx \sum_{k=1}^{K} \frac{\exp(r(z^k))}{\sum_{j=1}^{K} \exp(r(z^j))} \nabla_{x_t} r(z^k) \quad (13)$$

The proof is simple and follows and the reparameterization trick (Kingma & Welling, 2013) (see Section A.3). In Algorithm 1, we present how to perform guidance with Posterior Diamond Maps and in Algorithm 2 we present SMC. In the remainder of this section, we describe training of Posterior Diamond Maps in Section 4.1 and sampling in Section 4.2.

### 4.1. Training Posterior Diamond Maps via Distillation

We next discuss training of Posterior Diamond Maps. Flow maps are distilled ODE trajectories. Therefore, we first express sampling from $p_{1|t}$ via an ODE leveraging the re-

---

**Algorithm 1** Guidance with Posterior Diamond Maps

1: **Require:** Posterior Diamond Map $X_{s,s'}^{\theta}(\bar{x}_s|x_t, t)$, pre-trained flow model $u_t(x)$, $N$ simulation steps, $K$ Monte Carlo samples, differentiable reward $r$,
2: **Init:** $x_0 \sim \mathcal{N}(0, I_d)$
3: **Set** $h \leftarrow 1/N, t \leftarrow 0$
4: **for** $n = 0$ to $N - 1$ **do**
5:     **for** $k = 1$ to $K$ **do**
6:         $z^i \leftarrow X_{0,1}^{\theta}(\bar{x}_0^k|x_t, t) \quad (\bar{x}_0^k \sim \mathcal{N}(0, I_d))$
7:         $\nabla_{x_t} r(z^k) \leftarrow r(z^i).\text{backward}(x_t)$
8:     **end for**
9:     $\nabla_{x_t} V_t(x_t) \leftarrow \sum_k \text{softmax}(r(z^j)_{1 \leq j \leq K})[k] \nabla_{x_t} r(z^k)$
10:     $u_t^r \leftarrow u_t(x) + b_t \nabla_{x_t} V_t(x_t)$
11:     $X_{t+h} \leftarrow X_t + hu_t^r, \quad t \leftarrow t + h$
12: **end for**
13: **Return:** $X_1$

---

cently introduced GLASS Flows (Holderrieth et al., 2025). GLASS Flows samples from $p_{1|t}(\cdot|x_t)$ with conditional flow matching model where the condition is given by $x_t, t$ (i.e. it is fixed) and a new "inner" state $\bar{x}_s$ evolves from noise $\bar{x}_0$ to a sample $\bar{x}_1 \sim p_{1|t}(\cdot|x_t)$ with inner time $s$ ($0 \leq s \leq 1$). In our notation, we denote inner states with $\bar{x}$ while keeping $x$ for outer states. The **GLASS velocity field** is given by

$$\bar{u}_s(\bar{x}_s|x_t, t) \quad (\bar{x}_s, x_t \in \mathbb{R}^d, 0 \leq s, t \leq 1) \quad (14)$$

for which the sampling procedure is

$$X_0 \sim \mathcal{N}(0, I_d), \quad \frac{\mathrm{d}}{\mathrm{d}s} X_s = \bar{u}_s(\bar{x}_s|x_t, t) \quad (15)$$

$$\Rightarrow X_1 \sim p_{1|t}(\cdot|X_t = x_t), \quad (16)$$

which generates samples from the posterior $p_{1|t}$. Holderrieth et al. (2025) show how to obtain $\bar{u}_s(\bar{x}_s|x_t, t)$ from a pre-trained flow-matching model $u_t(x_t)$ via simple linear transformations of inputs and outputs. Specifically, define the **sufficient statistic** as the weighted average

$$S_{s,t}(\bar{x}_s, x_t) = \frac{\alpha_s \sigma_t^2 \bar{x}_s + \alpha_t \sigma_s^2 x_t}{\sigma_t^2 \alpha_s^2 + \alpha_t^2 \sigma_s^2}, \quad (17)$$

and the **time reparameterization** as:

$$t^*(s, t) = g^{-1}\left(\frac{\sigma_t^2 \sigma_s^2}{\sigma_t^2 \alpha_s^2 + \alpha_t^2 \sigma_s^2}\right), \quad g(t) = \frac{\sigma_t^2}{\alpha_t^2} \quad (18)$$

(see Section A.6 for analytical formula for $g^{-1}$). Then the GLASS velocity field is given by

$$\bar{u}_s(\bar{x}_s|x_t, t) = a_{s,t} \bar{x}_s + b_{s,t} D_{t^*}(\alpha_{t^*} S_{s,t}(\bar{x}_s, x_t)) \quad (19)$$

where $D_t$ is the denoiser (see (4)) and $a_{s,t}, b_{s,t} \in \mathbb{R}$ are weight coefficients (see Section A.4 for derivations and explicit formulae). Therefore, the above vector field is a linear

reparameterization of a pre-trained model in the denoiser parameterization. The intuition for the above formula is as follows: the sufficient statistics "summarizes" two states $\bar{x}_s, x_t$ into one state by taking their average. This "summary" is effectively a new state at a different time $t^*$ that we can denoise with the "old" denoiser (Holderrieth et al., 2025).

**Training.** To train a Posterior Diamond Flow Map model $X^\theta_{s,s'}(\bar{x}_s|x_t, t)$ with parameters $\theta$, one can now take an existing flow model $u_t(x)$, reparameterize to a GLASS velocity field $\bar{u}_s(\bar{x}_s|x_t, t)$, and then distill it with a distillation method of choice. For example, Lagrangian distillation would minimize the following loss (see equation (5)):

$$\mathbb{E}\left[\|\partial_{s'} X^\theta_{s,s'}(\bar{x}_s|x_t, t) - \text{sg}(u_{s'}(X_{s,s'}(\bar{x}_s|x_t, t)|x_t, t))\|^2\right]$$

where the expectation is over $z \sim p_{\text{data}}$, the times $s, s', t$ are sampled uniformly such that $s \leq s'$, and $x_t \sim p_t(\cdot|z), \bar{x}_s \sim p_s(\cdot|z)$.

*Remark* 4.2 (Training from scratch). Posterior Diamond Maps can also be trained from scratch via standard flow matching (see Section A.5). However, as distillation is much more efficient, e.g. can be done for large-scale models in several hours with 8 NVIDIA A100 GPUs (Sabour et al., 2025b), we focus on distillation in this work.

### 4.2. One-Step DDPM samples with Diamond Maps

Next, we discuss how one can *iteratively* sample from Posterior Diamond Maps. So far, we only discussed how to obtain terminal samples at time 1 via sampling from the posterior. However, to perform guidance, search, or SMC, we perform *iterative* step-wise sampling, i.e. we need a way to sample a state $x_{t'}$ given $x_t$ for $t' < 1$ that is not a terminal state.

A naive way of sampling from Posterior Diamond Maps would be via **iterative denoising and noising**, i.e. to sample from the posterior and then noise again

$$x_1 \leftarrow X^\theta_{0,1}(\bar{x}_0|x_t, t) \quad (\bar{x}_0 \sim \mathcal{N}(0, I_d)) \quad \text{(denoise)}$$
$$x_{t'} \leftarrow \alpha_{t'} x_1 + \beta_t \epsilon \quad (\epsilon \sim \mathcal{N}(0, I_d)) \quad \text{(noise)}$$

For a perfectly trained Posterior Diamond Map, this would be exact, i.e. $x_{t'} \sim p_{t'}$. However, in practice, flow maps have an error that increases with the simulation steps that are distilled in a flow map. Therefore, it seems suboptimal that the above sampling procedure goes from $t \rightarrow 1$ and back to $t'$ (e.g. set $t = 0.2, t' = 0.25$). Such an iterative denoising and noising scheme has already been shown to lead to error accumulation for flow maps and consistency models (Sabour et al., 2025b). As we demonstrate, a significantly better sampling procedure can be derived.

**Intuition.** The fundamental idea to improve this scheme relies on the following intuition: the Posterior Diamond Map has an inner time $s$ and an outer time $t$. By definition,

these are two *separate* time axis. However, intuitively $s = 0$ should correspond to outer time $t$ and $s = 1$ to outer time 1. But recall that our goal is to go to a time $t'$ in between $t$ and 1 ($t < t' < 1$)? Our approach is that we can find an inner time $s$ that corresponds to this $t'$ and thereby find a direct one step transition from $x_t$ to $x_{t'}$ in this way by mapping inner states $\bar{x}_s$ to outer states $x_t$ via the sufficient statistic.

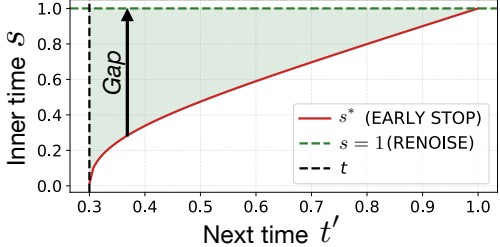

*Figure 2.* Effective time $0 \rightarrow s^*$ amortized in the flow map is significantly smaller for Diamond Early Stop DDPM sampling ("Early stop") than for iterative denoising and noising ("Renoise") leading to reduced error accumulation (see Figure 4). Here, we plot $s^*$ (red) as a function of $t, t'$ (see Section A.7).

To make this precise, we define **Diamond Early Stop DDPM sampling** as the sample that we obtain by stopping the Diamond flow "early" and re-transforming with the sufficient statistic $S_{s,t}$, i.e. for $\bar{x}_0 \sim \mathcal{N}(0, I_d)$ set

$$\bar{x}_{s^*} = X^\theta_{0,s^*}(\bar{x}_0|x_t, t) \quad \text{(flow stopped early)}$$
$$x_{t'} = \alpha_{t'} S_{s^*,t}(\bar{x}_{s^*}, x_t) \quad \text{(map inner to outer state)}$$

where we set $s^* = g^{-1}(g(t)g(t')/(g(t) - g(t')))$ with $g$ as in (18). The choice of $s^*$ is such that $t^*(s^*, t) = t'$ (see Section A.6). In the following, we call the **DDPM transitions** to be the transition probabilities $p^{\text{DDPM}}_{t'|t}(x_{t'}|x_t)$ from the time-reversal SDE (Ho et al., 2020; Song et al., 2020a) (see Section A.8). The DDPM transitions are a standard choice for reward alignment, in particular as a *proposal distribution* for SMC and search (Li et al., 2025b; Skreta et al., 2025; Singhal et al., 2025). We present:

**Proposition 4.3** (Diamond DDPM sampling). *Let a time $t' > t$ and $p^{\text{DDPM}}_{t'|t}$ be the DDPM transition kernel. Then for $x_{t'}$ obtained via Diamond DDPM sampling:*

$$x_{t'} \sim p^{\text{DDPM}}_{t'|t}(\cdot|x_t)$$

A proof can be found in Section A.8. The above proposition is remarkable, as it shows that the DDPM flow map is contained in the Posterior Diamond Map, although the model was not trained for it. Therefore, Posterior Diamond Map not only accelerates value function estimation but also unlocks more efficient proposal distribution for SMC and search via Diamond DDPM Sampling. As we will demonstrate in Section 7.1, Diamond Early Stop DDPM sampling leads to significantly improved performance compared to iterative denoising and noising.

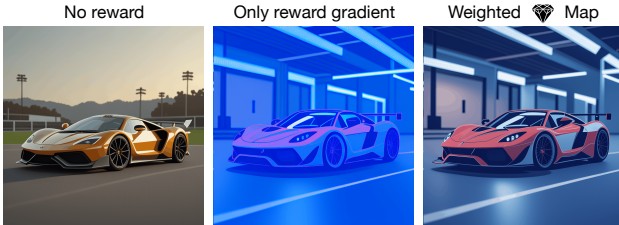

No reward | Only reward gradient | Weighted 💎 Map

*Figure 3.* Illustration of Theorem 5.1 with a blueness reward (i.e. reward is maximized for full blue image). Left: Sampling from SANA-Sprint model with no reward. Middle: Naive re-noising and reward gradient with no weighting ((22)). The entire image becomes blue close to collapse. Right: Corrected gradient value function via Theorem 5.1: image remains more realistic due to the added regularization preventing drift off the data manifold.

## 5. Weighted Diamond Maps

In this section, we propose a different approach to value function estimation via flow maps. We ask: can we estimate the value function by using an existing "standard" flow map $X_{t,t'}$? As we show in this section, this is indeed possible.

Let $X_{t,t'}$ be a normal flow map. For a given $x_t \in \mathbb{R}^d$, the standard flow map $X_{t,t'}(x_t)$ is deterministic. However, a natural idea is to **renoise** $x_t$ back to an earlier time $x_{t'}$ for $t' < t$ and use the flow map to map it to time 1. Specifically, define the **renoising map** for $t' < t$ as

$$x_{t'}(x_t, \epsilon) = \frac{\alpha_{t'}}{\alpha_t} x_t + \left( \sqrt{\sigma_{t'}^2 - \frac{\alpha_{t'}^2}{\alpha_t^2} \sigma_t^2} \right) \epsilon \quad (20)$$

The renoising map is constructed such that if $\epsilon \sim \mathcal{N}(0, I_d)$, then $x_{t'}(x_t, \epsilon)$ corresponds to simulation of the diffusion forward process from $t$ to $t'$ (Song et al., 2020b; Sohl-Dickstein et al., 2015). We now use this renoising map to make the flow map stochastic. For $\epsilon \sim \mathcal{N}(0, I_d)$, we set

$$x_{t'} = x_{t'}(x_t, \epsilon) \qquad \text{(renoise)} \quad (21)$$
$$z(x_t, \epsilon) = X_{t',1}(x_{t'}) \qquad \text{(map to data)} \quad (22)$$

This results in a distribution $z(x_t, \epsilon) \sim q_{1|t}(\cdot|x_t)$ over "clean" data that intuitively should sample "locally" around $x_t$. However, the resulting distribution is *not* the posterior distribution, i.e. $q_{1|t}(\cdot|x_t) \neq p_{1|t}(\cdot|x_t)$. Therefore, unlike the direct distillation approach in the previous section, samples from this distribution would *not* lead to a correct estimator of the value function. This is illustrated in Figure 3.

However, our fundamental idea is that one correct for it by a *recovery reward* defined via:

$$r_{\text{recov}}(x_t, \epsilon) := r(z(x_t, \epsilon)) - \frac{\|x_t - \alpha_t z(x_t, \epsilon)\|^2}{2\sigma_t^2} \quad (23)$$

The second summand is a recovery reward that is high if $x_t$ is a plausible noisy version of the suggested clean data point $z(x_t, \epsilon_i)$. It equals the NLL $-\log p_t(x|z)$ of the probability path. Refining this intuition, we establish:

**Proposition 5.1** (Weighted Diamond Map). *For every $k = 1, \cdots, K$ let $\epsilon_k \sim \mathcal{N}(0, I_d)$. Then the following **weighted Diamond guidance estimator** is a consistent estimator of the gradient of the value function*

$$\nabla_{x_t} V_t(x_t) \approx \sum_{i=k}^{K} w_k \left[ \nabla_{x_t} r_{\text{recov}}(x_t, \epsilon_k) + \delta_{\text{score}}^k \right],$$

*where $\delta_{\text{score}}^k = \nabla \log p_{t'}(x_{t'}^k) \frac{\alpha_{t'}}{\alpha_t} - \nabla \log p_t(x_t)$ is defined as a score-based correction and the weight coefficients $w_i$ are given by $w_k = \text{softmax}(v_1, \cdots, v_N)$ and*

$$v_k = r_{\text{recov}}(x_t, \epsilon_k) + \gamma_k + \frac{1}{2} \|\epsilon_k\|^2$$

$$\gamma_k = (x_{t'}^k - x_t)^T \int_0^1 \nabla \log p_{t'}(x_t + u(x_{t'}^k - x_t)) \mathrm{d}u$$

We present a proof in Section A.9 and the method in Algorithm 3. We show in Figure 3 that the additional terms are crucial for exact guidance estimation. The estimator is tractable as we can compute $\delta_{\text{score}}$ because the score function $\nabla \log p_t(x)$ is a reparameterization of the velocity field $u_t(x)$. As $t'$ is close to $t$, the $\gamma$-coefficients can be approximated efficiently with a trapezoidal quadrature

$$\gamma_k \approx \frac{1}{2} \left[ \nabla \log p_{t'}(x_t) + \nabla \log p_{t'}(x_{t'}^k) \right]^T (x_{t'}^k - x_t)$$

The above estimator can be computed in $\mathcal{O}(2KN)$ function evaluations where $K$ is the number of Monte Carlo samples and $N$ the number of simulation steps of the ODE (one call of the flow map $X_{t',1}$ for $r_{\text{recov}}$ and one call of the flow model to obtain $\delta_{\text{score}}$ per Monte Carlo sample). To reduce it further, we can use that $\nabla \log p_{t'}(x_{t'}) = (\alpha_{t'} D_{t'}(x_{t'}) - x_{t'})/\sigma_{t'}^2$ by Tweedie's formula (Efron, 2011). We can approximate this via the flow map by setting the denoiser equal to the flow map $D_t(x_t) \approx X_{t,1}(x_t)$ and use

$$\nabla \log p_{t'}(x_{t'}) \approx \frac{\alpha_{t'} X_{t',1}(x_{t'}) - x_{t'}}{\sigma_{t'}^2} \quad (24)$$

This reduces the number of function evaluations to $\mathcal{O}(KN)$ (as only the flow map is called per Monte Carlo sample). Further, it is well-known that the interval where diffusion models generate the semantically meaningful part happens in a narrow *critical window* (Luo et al., 2023; Park et al., 2023) We leverage this fact and only perform guidance in a critical guidance window $0 \leq t_{\text{guid-min}} \leq t_{\text{guid-max}} \leq 1$. Therefore, the total number of function evaluations amounts to $\mathcal{O}(K * N_{\text{guidance}} + N_{\text{no-guidance}})$ where $N_{\text{no-guidance}}$ are the steps where no guidance is performed. We summarize the method in Algorithm 3. Finally, we note that the Weighted Diamond Map can also be used to estimate the value function itself (and not only its gradient), see Section A.10.

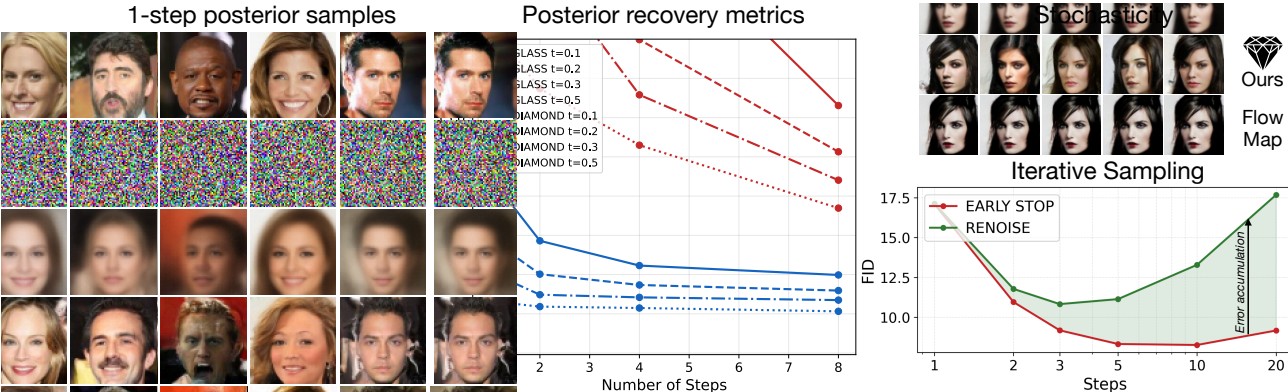

*Figure 4.* **Training and Sampling Posterior Diamond Maps.** Left: Example of one-step posterior samples from Posterior Diamond Maps. The one-step samplers are faithful and of high quality. Middle: Quantitative results for posterior sampling for various times $t$. Posterior Diamond Maps outperforms GLASS Flows and therefore have successively distilled them. Top right: In contrast to flow maps that always return the same sample, Posterior Diamond Maps are stochastic and allow to explore future possibilities (sample from posterior). Bottom right: Iterative sampling leads to error accumulation via a iterative denoising and noising scheme (Section 4.2), while improves for Diamond Early Stop DDPM sampling. All results are for CelebA-64 dataset.

**Posterior Diamond Map vs. Weighted Diamond Map.** We finally discuss the advantages and disadvantages of the Posterior vs. Weighted Diamond Map for both training and inference cost. For training, the Posterior Diamond Map requires distillation of GLASS Flows, while the Weighted Diamond Map requires only to distill a standard flow map. Practically speaking, this means that for a practitioner many models can be used off-the-shelf for Weighted Diamond Maps already. For inference cost, it is instructive to think of both Posterior Diamond Maps and Weighted Diamond Maps as effectively estimating the value function $V_t(x_t)$ via importance sampling (i.e. via reweighting a proposal distribution). The accuracy of such estimators depend on the *effective sample size* (Casella & Berger, 2024). Note the ESS is maximal (highest efficiency) if the importance weights are constant. For the Posterior Diamond Maps, the importance sampling weights are $\exp(r(z))$, while for the Weighted Diamond Maps it consists of several terms (see Theorem 5.1). We anticipate that for *perfect training* Posterior Diamond Maps offers more efficiency at inference time. We consider it an open theoretical question to study the merits of each method rigorously and anticipate that its answer depends on the reward $r(z)$.

## 6. Related Work

We discuss most closely related work in this section and refer to Section C for an extended discussion. There has been recent interest in using flow maps and distilled diffusion models for reward alignment. Eyring et al. (2024) leverage a one-step model to optimize the initial noise sample $x_0$ with gradient descent methods (Ben-Hamu et al., 2024; Wang et al., 2025b). Sabour et al. (2025a) use flow maps to obtain efficient one-step look-aheads and replace the denoiser ap-

proximation (see (10)) with a flow map approximation that can be corrected for via importance sampling. Our focus here is on estimating the value function, an arguably harder estimation task. Concurrently to our work, Potaptchik et al. (2026) also study stochastic flow maps to sample from the posterior distribution $p_{1|t}$ for reward alignment, i.e. here called *Posterior Diamond Maps*. We additionally propose *Weighted Diamond Maps* (see Section 5) reusing existing flow maps and Diamond DDPM sampling. Other works have explored learning approximations of the posterior $p_{1|t}$ beyond just learning the mean via the denoiser (Elata et al., 2024; De Bortoli et al., 2025; Chen et al., 2025a). We note that our method is not approximate but exact.

## 7. Experiments

### 7.1. Posterior Diamond Maps

We present experiments for Posterior Diamond Maps. We distill Posterior Diamond Flow models from pre-trained models on CIFAR10 (Krizhevsky, 2009) and CelebA (Liu et al., 2015) (unconditional) and ImageNet1k 256x256 (Deng et al., 2009) (class-conditional). To test the ability to sample from the posterior $p_{1|t}$, we follow a similar setup as in (Holderrieth et al., 2025). We present key qualitative and quantitative results in Figure 4, Figure 12, Figure 5. For comprehensive experimental results, reference Section D.1.

As one can see in Figure 4, Diamond Maps allows us to sample from the posterior in one step, distilling GLASS Flows effectively into a one-step sampler. We also test Diamond DDPM sampling (see Theorem 4.3) vs. standard iterative noising and renoising (see Section 4.2). Diamond Early Stop DDPM sampling outperforms iterative denoising and noising significantly, making it the method of choice for

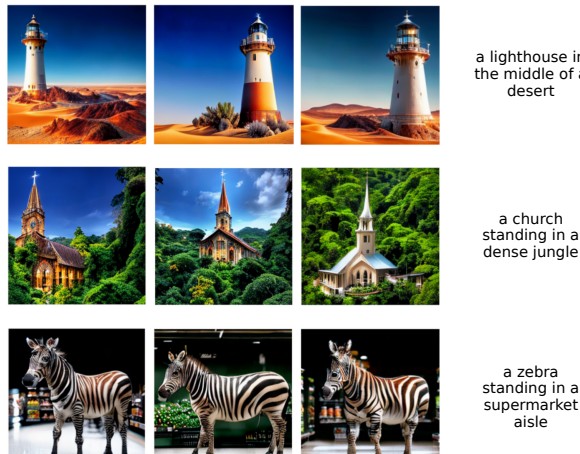

a lighthouse in
the middle of a
desert

a church
standing in a
dense jungle

a zebra
standing in a
supermarket
aisle

*Figure 5.* Making a class-conditional ImageNet model text-conditioned using Posterior Diamond Maps. Qualitative examples of ImageReward-reward alignment with guidance with Posterior Diamond Maps. Model trained on ImageNet1k (class-conditional). Model generates true out-of-distribution images following the prompts. More examples in Figure 19.

iterative sampling (see Figure 4 and Table 4). We present more quantitative results for few step sampling and posterior sampling in Table 2 and Table 3. Overall, these show that a Posterior Diamond Maps model can be successfully trained via standard flow map distillation.

We also apply guidance with Posterior Diamond Maps to common noisy linear inverse problems and prompt alignment (see Figure 13 and Figure 19 for examples). As a baseline, we use standard guidance via the denoiser approximation (see (10)) commonly called *Diffusion Posterior Sampling (DPS)* (Chung et al., 2022). As an additional baseline, we use a flow map approximation as $V_t(x_t) \approx r(X_{t,1}(x_t))$, e.g. used in (Sabour et al., 2025b). This can be considered a version of Weighted Diamond Maps for $t' = t$. For inverse problems, we observe that the Posterior Diamond map has a **better Pareto frontier** over various reward scales and NFEs (Figure 12). In the case of CelebA-64, we find Posterior Diamond Maps is more robust towards high reward scales, caused by stochasticity, compared to guidance with a naive flow map approximation (see Figure 8). For ImageNet, we find that scaling base drift steps quickly reaches diminishing returns and allocating these NFEs to Monte Carlo samples help continue scaling (see Figure 12). We also present experiments making ImageNet/CelebA models text-conditioned using text-image alignment reward models (Radford et al., 2021; Xu et al., 2023)using SMC (see Figure 10) and reward guidance (see Figure 19). As one can in Figure 5, Posterior Diamond Maps have a strong ability to guide the model to out-of-distribution images that would be virtually impossible to be sampled without guidance (i.e. just by a Best-of-N scheme). This effectively realizes the promise of stochastic exploration enabled by stochastic flow maps.

## 7.2. Weighted Diamond Maps

**High-resolution text-to-image generation.** We next evaluate the effectiveness of *Weighted Diamond Maps* applied to a pre-trained flow map *without* any fine-tuning, in the setting of text-to-image generation with human-preference reward alignment. We consider two base models operating at high resolution: SANA-Sprint 0.6B (Chen et al., 2025b) and a flow-map distillation of FLUX.1-dev[2]. We follow Eyring et al. (2024) and leverage a linear combination of four pre-trained human-preference reward models. As a disentangled evaluation from these rewards, we primarily benchmark improvements on GenEval (Ghosh et al., 2023) and UniGenBench++ (Wang et al., 2025a) to ensure no reward hacking.

**Analysis on FLUX.** We begin with the FLUX Flow Map model ($n$=25 ODE steps), and compare Diamond Maps against two primary baselines: Best-of-$N$ (Karthik et al., 2023; Ma et al., 2025), which generates $N$ independent samples and returns the highest-reward one, and Flow Map Guidance (Sabour et al., 2025b). We ablate three key hyperparameters: the number of guidance steps $g$, the perturbation noise level (parameterized by its signal-to-noise ratio, SNR; Section D.2), and the number of Monte Carlo samples mc. Figure 16 reports GenEval scores across guidance and noise configurations. Diamond Maps consistently improve over the base model, Best-of-$N$, and Flow Map Guidance (Sabour et al., 2025b) in all settings. The noise level plays a critical role: SNR=1.5 yields the strongest results, while insufficient noise (SNR=1.1) limits the estimator's exploration and excessive noise (SNR$\geq$3.0) degrades quality—particularly when guidance is applied at many steps ($g$=24), where perturbation errors accumulate along the ODE trajectory. Most notably, increasing the number of Monte Carlo samples mc yields consistent gains. A moderate guidance frequency ($g$=10) offers the best trade-off between alignment strength and trajectory coherence; we fix this configuration for all subsequent FLUX experiments.

**Compute scaling on FLUX.** Diamond Maps admit a scaling axis fundamentally different from existing methods: rather than drawing independent samples (Best-of-$N$) or refining the ODE discretization (Flow Map Guidance), we increase the number of Monte Carlo particles mc used to approximate the reward-tilted velocity at each guidance step. The total cost is $n + 2 * g * $mc NFEs for Diamond Maps, $n * N$ for Best-of-$N$, and $n + g$ for Flow Map Guidance, where we scale its number of inference-plus-guidance steps as this is its most natural compute axis (scaling other quantities such as particles would equally benefit Diamond Maps, confounding the comparison). Figures 6a and 6b compare all three strategies as a function of total NFEs on GenEval

---

[2]https://huggingface.co/gabeguofanclub/
flux-1-dev-flowmap-lsd

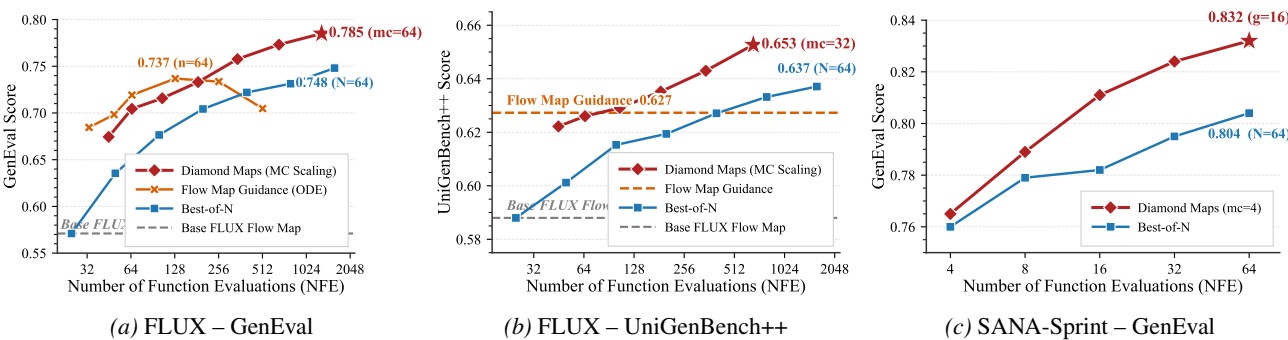

*Figure 6.* Compute scaling (NFEs) for Diamond Maps, Best-of-$N$, and Flow Map Guidance across models and benchmarks. Diamond Maps (MC scaling) consistently achieve a steeper Pareto frontier than Best-of-$N$ (independent samples) and Flow Map Guidance (additional ODE steps), confirming more favorable compute–quality trade-offs across both velocity-parameterized (FLUX) and direct-prediction (SANA-Sprint) flow maps.

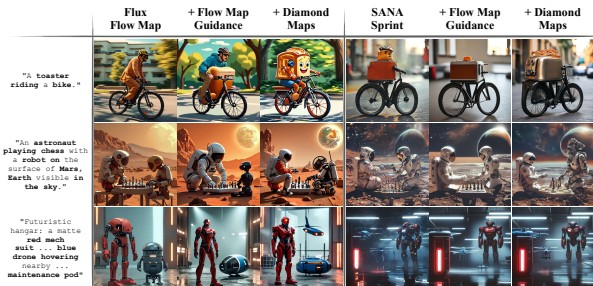

*Figure 7.* Qualitative comparison on compositional prompts for FLUX Flow Map and SANA-Sprint, same seed per row.

and UniGenBench++, both of which are disentangled from the reward models used during guidance. On GenEval, Diamond Maps reach 0.785 at mc=64, outperforming both Flow Map Guidance at its peak (0.737, after which performance *declines* with additional steps) and Best-of-$N$ (0.748 at $N$=64). On UniGenBench++, the picture is consistent: We employ the best-performing parameters from GenEval with which Diamond Maps achieves 0.653 versus 0.637 for Best-of-$N$, while Flow Map Guidance achieves 0.627 at $g = 24, n = 25$. Crucially, Diamond Maps exhibit qualitatively different scaling behavior: while Best-of-$N$ faces diminishing returns from selecting among independent samples and Flow Map Guidance saturates or even degrades with additional steps, Diamond Maps steadily improve as mc grows by tightening the Monte Carlo approximation.

**Results on SANA-Sprint.** SANA-Sprint presents a fundamentally different setting from FLUX: as a consistency-distilled model, it directly predicts the clean image $x_1$ at each step rather than a velocity field, yielding strong single-step generations but lacking an explicit velocity parameterization. Table 6 compares Diamond Maps against Best-of-$N$ (Karthik et al., 2023; Ma et al., 2025), Reward-based Noise Optimization (Eyring et al., 2024), and Prompt Optimization (Mañas et al., 2024): Weighted Diamond Maps outperform competing approaches while using fewer NFEs,

directly fulfilling the promise of efficient test-time reward alignment. Figure 6c further traces the Pareto frontier against Best-of-$N$ as compute grows: Diamond Maps scale more favorably. Together with the FLUX results, these experiments confirm that Diamond Maps provide a unified, compute-efficient mechanism for reward-guided generation consistently delivering better scaling than alternatives.

## 8. Discussion

We introduced *Diamond Maps*, a framework to enable efficient and accurate alignment of flow and diffusion models at inference-time using flow maps. We present two methods: *Posterior Diamond Maps* demonstrate that one can distill GLASS Flows into one-step samplers for the posterior itself, while *Weighted Diamond Maps* show that even standard off-the-shelf flow maps can be turned into consistent value-function estimators through a renoising procedure.

We note two limitations. First, although our estimators are asymptotically accurate (consistent) and efficient, they could struggle with arbitrary reward-alignment problems: when the support of $p_{\text{data}}$ is far from that of the reward-tilted distribution $p^r(z)$, the variance of estimators can become very large. This is not specific to Diamond Maps but reflects a fundamental difficulty of inference-time alignment when the prior has weak coverage of high-reward regions. Second, the practical accuracy of all estimators depends on the quality of the underlying flow maps; while sufficiently accurate maps should in principle preserve one-step sampling performance, current distilled models still exhibit a gap relative to their teachers, which we expect continued progress to close. While our experiments focus on images, the framework applies directly to any modality where flow or diffusion models are used. We view Diamond Maps as a significant step toward generative models that act as general-purpose inference engines, capable of optimizing for arbitrary user preferences and constraints on demand.

## Acknowledgements

Peter Holderrieth thanks Brian Karrer, Yaron Lipman, Ricky T.Q. Chen, Uriel Singer, Karsten Kreis, Brian Trippe, and Julius Berner for helpful discussions and helpful feedback on early drafts of the work. Peter Holderrieth acknowledges support from the Machine Learning for Pharmaceutical Discovery and Synthesis (MLPDS) consortium and the NSF Expeditions grant (award 1918839) Understanding the World Through Code. Max Simchowitz acknowledges funding from a TRI U2.0 grant. Zeynep Akata acknowledges funding by the ERC (853489 - DEXIM) and the Alfried Krupp von Bohlen und Halbach Foundation. Luca Eyring would like to thank the European Laboratory for Learning and Intelligent Systems (ELLIS) PhD program for support. Luca Eyring is supported by the Google PhD Fellowship in Machine Learning.

## Impact Statement

This paper presents work whose goal is to advance the field of Machine Learning. There are many potential societal consequences of our work, none which we feel must be specifically highlighted here.

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

# A. Mathematical Details and Proofs

## A.1. Details for background on flow matching

The conditional vector field is given by (Lipman et al., 2022; 2024):

$$u_t(x|z) = \left( \dot{\alpha} - \frac{\dot{\beta}_t}{\beta_t} \alpha_t \right) z + \frac{\dot{\beta}_t}{\beta_t} x \tag{25}$$

## A.2. Derivation of Value Function and Proof of (9)

*Proof.* We can derive:

$$p_t^r(x) = \frac{1}{Z} \int p_t(x|z) \exp(r(z)) p_{\text{data}}(z) \mathrm{d}z \tag{26}$$

$$\Rightarrow \quad \frac{p_t^r(x)}{p_t(x)} = \frac{1}{Z} \int \exp(r(z)) \underbrace{\frac{p_t(x|z) p_{\text{data}}(z)}{p_t(x)}}_{\text{posterior}} \mathrm{d}z = \frac{1}{Z} \mathbb{E}[\exp(r(z))|x_t = x] = \frac{1}{Z} \exp(V_t(x)) \tag{27}$$

This shows (8). We can express the conditional and marginal vector field via the conditional and marginal score function (Lipman et al., 2024):

$$u_t(x|z) = a_t x + b_t \nabla \log p_t(x|z)$$
$$u_t(x) = a_t x + b_t \nabla \log p_t(x)$$
$$\text{where} \quad a_t = \frac{\dot{\alpha}_t}{\alpha_t}, b_t = \beta_t^2 \frac{\dot{\alpha}_t}{\alpha_t} - \dot{\beta}_t \beta_t$$

As this holds for any data distribution, it also holds for the reward-tilted distribution. Therefore, the tilted marginal vector field $u_t^r$ is given by

$$u_t^r(x) = a_t x + b_t \nabla \log p_t^r(x) = u_t(x) + b_t \nabla \log \frac{p_t^r(x)}{p_t(x)} \tag{28}$$

Taking the gradient of the $\log$ and inserting it into (28), we obtain:

$$u_t^r(x) = u_t(x) + b_t \nabla_{x_t} V_t(x_t)$$

This finishes the proof. $\qquad \square$

## A.3. Proof of Theorem 4.1

*Proof.* Recall that the value function is given by

$$V_t(x_t) = \log \mathbb{E}_{z \sim p_{1|t}(\cdot|x_t)}[\exp(r(z))] \tag{29}$$

Therefore, for $z^1, \cdots, z^N \sim p_{1|t}(\cdot|x_t)$:

$$V_t(x_t) \approx \log \frac{1}{N} \sum_{i=1}^{N} \exp(r(z^i)) \tag{30}$$

is a consistent estimator of the value function. Setting $z^i = X_{0,1}(\bar{x}_0|x_t, t)$ proves that (12) is a consistent estimator.

Further, by differentiating (12) by $x_t$ we obtain:

$$\nabla_{x_t} V_t(x_t) \approx \frac{1}{\frac{1}{N}\sum_{i=1}^{N} \exp(r(z^i))} \nabla_{x_t} \frac{1}{N} \sum_{i=1}^{N} \exp(r(z^i)) \tag{31}$$

$$= \frac{1}{\sum_{i=1}^{N} \exp(r(z^i))} \sum_{i=1}^{N} \nabla_{x_t} \exp(r(z^i)) \tag{32}$$

$$= \frac{1}{\sum_{i=1}^{N} \exp(r(z^i))} \sum_{i=1}^{N} \exp(r(z^i)) \nabla_{x_t} r(z^i) \tag{33}$$

$$= \sum_{i=1}^{N} \frac{\exp(r(z^i))}{\sum_{i=1}^{N} \exp(r(z^i))} \nabla_{x_t} r(z^i) \tag{34}$$

This shows (13). $\qquad\qquad\square$

### A.4. GLASS velocity field for posterior

The derivation follows (Holderrieth et al., 2025) for the special case of transition being equal to the posterior $p_{1|t}$. First, we set the following parameters in (Holderrieth et al., 2025, Theorem 1)

$$t' = 1, \rho = 0, \bar{\alpha}_s = \alpha_s, \bar{\sigma}_s = \sigma_s$$

and obtain - using their notation

$$\bar{\gamma} = 0, \mu(s) = \begin{pmatrix} \alpha_t \\ \bar{\alpha}_s \end{pmatrix}, \Sigma(s) = \begin{pmatrix} \sigma_t^2 & 0 \\ 0 & \bar{\sigma}_s^2 \end{pmatrix}$$

$$w_1(s) = \dot{\sigma}_s/\sigma_s, w_2(s) = \dot{\alpha}_s - \alpha_s \dot{\sigma}_s/\sigma_s, w_3(s) = 0$$

The sufficient statistics $S_{s,t}(\bar{x}_s, x_t)$ is then given by

$$S_{s,t}(\bar{x}_s, x_t) \tag{35}$$

$$= \frac{\mu^T \Sigma^{-1} [x_t, \bar{x}_s]}{\mu^T \Sigma^{-1} \mu} \tag{36}$$

$$= \frac{\frac{\alpha_s}{\sigma_s^2} \bar{x}_s + \frac{\alpha_t}{\sigma_t^2} x_t}{\frac{\alpha_s^2}{\sigma_s^2} + \frac{\alpha_t^2}{\sigma_t^2}} \tag{37}$$

$$= \frac{\alpha_s \sigma_t^2 \bar{x}_s + \alpha_t \sigma_s^2 x_t}{\sigma_t^2 \alpha_s^2 + \alpha_t^2 \sigma_s^2}, \tag{38}$$

and the **time reparameterization** as:

$$t^*(s,t) = g^{-1}\left(\frac{1}{\mu^T \Sigma^{-1} \mu}\right) = g^{-1}\left(\frac{1}{\frac{\alpha_s^2}{\sigma_s^2} + \frac{\alpha_t^2}{\sigma_t^2}}\right) = g^{-1}\left(\frac{\sigma_t^2 \sigma_s^2}{\sigma_t^2 \alpha_s^2 + \alpha_t^2 \sigma_s^2}\right), \quad g(t) = \frac{\sigma_t^2}{\alpha_t^2} \tag{39}$$

Therefore,

$$D_{\mu(s),\Sigma(s)}(x_t, \bar{x}_s) = D_{t^*}(\alpha_{t^*} S_{s,t}(\bar{x}_s, x_t))$$

Hence, for these choices (Holderrieth et al., 2025, Equation 14) implies that

$$\bar{u}_s(\bar{x}_s | x_t, t) = w_1(s)\bar{x}_s + w_2(s) D_{t^*}(\alpha_{t^*} S_{s,t}(\bar{x}_s, x_t))$$

which coincides with (19) after setting $a_{s,t} = w_1(s) = \dot{\sigma}_s/\sigma_s$ and $b_{s,t} = w_2(s) = \dot{\alpha}_s - \alpha_s \dot{\sigma}_s/\sigma_s$.

## A.5. Training Posterior Diamond Maps from scratch

In the same way as for distillation from GLASS Flows, any self-distillation loss could be used for training Posterior Diamond Maps from scratch. To illustrate this, let $X^\theta_{s,s'}(\bar{x}_s|x_t, t) = \bar{x}_s + (s' - s)v^\theta_{s,s'}(x|x_t, t)$ be a flow map parameterized by an average velocity field $v^\theta_{s,s'}(x)$. Then we can train it via Lagrangian or Eulerian self-distillation (Boffi et al., 2025b)

$$L_{\text{Lag}}(\theta) = \mathbb{E}\left[\|v^\theta_{s,s}(\bar{x}_s|x_t, t) - u_s(\bar{x}_s|z)\|^2\right] + \mathbb{E}\left[\|\partial_{s'}X^\theta_{s,s'}(\bar{x}_s|x_t, t) - v^\theta_{s',s'}(X^\theta_{s,s'}(\bar{x}_s|x_t, t)|x_t, t)\|^2\right] \quad (40)$$

$$L_{\text{Eul}}(\theta) = \mathbb{E}\left[\|v^\theta_{s,s}(\bar{x}_s|x_t, t) - u_s(\bar{x}_s|z)\|^2\right] + \mathbb{E}\left[\|\partial_s X^\theta_{s,s'}(\bar{x}_s|x_t, t) + \nabla_{\bar{x}_s} X^\theta_{s,s'}(\bar{x}_s|x_t, t)v_{s,s'}(\bar{x}_s|x_t, t)\|^2\right] \quad (41)$$

where the expectation is over $z \sim p_{\text{data}}$, the times $s, s', t$ are sampled uniformly such that $s \leq s'$, and $x_t \sim p_t(\cdot|z), \bar{x}_s \sim p_s(\cdot|z)$, and $u_s(\bar{x}_s|z)$ is the conditional vector field (see (25)).

## A.6. Analytical formulas for $g^{-1}$

We derive specific formulas for $g, g^{-1}$ for various schedulers $\alpha_t, \sigma_t$. This is stated here for completeness (Holderrieth et al., 2025).

**Linear schedule.** Let us set

$$\alpha_t = t, \quad \sigma_t = 1 - t$$

Then:

$$g(t) = \frac{(1-t)^2}{t^2} = \left(\frac{1}{t} - 1\right)^2$$

$$g^{-1}(y) = \frac{1}{1 + \sqrt{y}}$$

**Variance-preserving schedule.** Let us set:

$$\sigma_t = \sqrt{1 - t}, \quad \alpha_t = \sqrt{t}$$

Then:

$$g(t) = \frac{1 - t}{t} = \frac{1}{t} - 1$$

$$g^{-1}(y) = \frac{1}{1 + y}$$

**Variance-exploding schedule.** Let us set:

$$\sigma_t = \sqrt{t}, \quad \alpha_t = 1$$

Then:

$$g(t) = t$$

$$g^{-1}(y) = \frac{1}{y}$$

## A.7. Derivation of $s^*$

Recall the definitions:

$$g(t) = \frac{\sigma_t^2}{\alpha_t^2}$$

$$t^*(s, t) = g^{-1}\left(\frac{\sigma_t^2 \sigma_s^2}{\alpha_s^2 \sigma_t^2 + \alpha_t^2 \sigma_s^2}\right) = g^{-1}\left(\frac{1}{\frac{1}{g(t)} + \frac{1}{g(s)}}\right)$$

$$s^*(t, t') = g^{-1}\left(\frac{g(t)g(t')}{g(t) - g(t')}\right)$$

As described in the main text, $s^*$ was chosen such that $t^*(s^*, t) = t'$. To prove this, we simply plugin the equations:

$$t^*(s^*(t, t'), t) = g^{-1}\left(\frac{1}{\frac{1}{g(t)} + \frac{g(t)-g(t')}{g(t)g(t')}}\right)$$

$$= g^{-1}\left(\frac{1}{\frac{g(t)}{g(t)g(t')}}\right)$$

$$= g^{-1}\left(g(t')\right)$$

$$= t'$$

### A.8. Formal definition of DDPM transitions and proof of Theorem 4.3

The forward process (Sohl-Dickstein et al., 2015; Ho et al., 2020; Song et al., 2020b) is defined via a transition kernel $q_{t|t'}$ in reverse-time given by:

$$q_{t|t'}(x_t|x_{t'}) = \mathcal{N}\left(\frac{\alpha_t}{\alpha_{t'}}x_{t'}; \left(\sigma_t^2 - \frac{\alpha_t^2}{\alpha_{t'}^2}\sigma_{t'}^2\right)I\right), \quad (t < t')$$

The time-reversal SDE or DDPM transitions (Song et al., 2020b; Sohl-Dickstein et al., 2015; Ho et al., 2020) are then given by

$$p_{t'|t}^{\text{DDPM}}(x_{t'}|x_t) = q_{t|t'}(x_t|x_{t'})\frac{p_{t'}(x_{t'})}{p_t(x_t)}$$

*Proof of Theorem 4.3.* For $\bar{x}_0 \sim \mathcal{N}(0, I_d)$, we define the random variable

$$\tilde{X}_{t'} = \alpha_{t'}S_{s^*,t}(X_{0,s^*}(\bar{x}_0|x_t, t))$$

Our goal is to show that

$$\tilde{p}(\tilde{X}_{t'} = x_{t'}|X_t = x_t) = p_{t'|t}^{\text{DDPM}}(X_{t'} = x_{t'}|X_t = x_t)$$

i.e. that the distribution of $\tilde{X}_{t'}$ given $X_t = x_t$ coincides with the transition kernel $p_{t'|t}^{\text{DDPM}}$ of the DDPM transitions. To show this, it is sufficient to show that the joint distributions coincide

$$\tilde{p}(\tilde{X}_{t'} = x_{t'}, X_t = x_t) = p_{t',t}(X_{t'} = x_{t'}, X_t = x_t) \tag{42}$$

for $x_t \sim p_t$. As the DDPM transition of $q_{t'|t}$, we know that:

$$p_{t,t'}(x_t, x_{t'}) = \int q_{t|t'}(x_t|x_{t'})p_{t'}(x_{t'}|x_1)p_{\text{data}}(x_1)\mathrm{d}x_1$$

$$\text{where} \quad q_{t|t'}(x_t|x_{t'}) = \mathcal{N}\left(\frac{\alpha_t}{\alpha_{t'}}x_{t'}; (\sigma_t^2 - \frac{\alpha_t^2}{\alpha_{t'}^2}\sigma_{t'}^2)I\right), p_{t'}(x_{t'}|x_1) = \mathcal{N}(\alpha_{t'}x_1; \sigma_{t'}^2 I)$$

In other words, we get a sample $(X_t, X_{t'}) \sim p_{t,t'}(x_t, x_{t'})$ via

$$X_1 \sim p_{\text{data}},$$
$$X_{t'} = \alpha_{t'}X_1 + \sigma_{t'}\epsilon_1,$$
$$X_t = \frac{\alpha_t}{\alpha_{t'}}X_{t'} + \sqrt{\sigma_t^2 - \frac{\alpha_t^2}{\alpha_{t'}^2}\sigma_{t'}^2}\epsilon_2 \quad \epsilon_1, \epsilon_2 \sim \mathcal{N}(0, I_d)$$

Next, to sample from $\tilde{p}(\tilde{x}_t, x_t)$, we know that we can equivalently do the following procedure: Sample $X_1 \sim p_{\text{data}}$ and then noise $X_t$ and $X_{s*}$ independently and map $X_{s*}$ to $X_{t'}$. In other words, we can use the following sampling procedure:

$$X_1 \sim p_{\text{data}}, \tag{43}$$

$$X_{s*} = \bar{\alpha}_{s*} X_1 + \bar{\sigma}_{s*} \tilde{\epsilon}_1 \tag{44}$$

$$X_t = \alpha_t X_1 + \sigma_t \tilde{\epsilon}_2 \tag{45}$$

$$X_{t'} = \alpha_{t'} \frac{\bar{\alpha}_{s*} \sigma_t^2 X_{s*} + \alpha_t \bar{\sigma}_{s*}^2 X_t}{\bar{\alpha}_{s*}^2 \sigma_t^2 + \alpha_t^2 \bar{\sigma}_{s*}^2} \tag{46}$$

This is because of the definition of the GLASS velocity field (Holderrieth et al., 2025) and GLASS probability path (Holderrieth et al., 2025, Section 4.2.2). We now only have to show that $(X_t, X_{t'})$ sampled via this procedure has the same distribution as sampling them via the SDE/DDPM procedure. However, for that, it is sufficient to show that their distribution conditioned on $X_1$ is equal. Therefore, let's fix $X_1 = x_1$. Then for the DDPM/SDE case, it holds that

$$(X_t, X_{t'})|X_1 = x_1 \sim \mathcal{N}\left(\begin{pmatrix} \alpha_t x_1 \\ \alpha_{t'} x_1 \end{pmatrix}, \begin{pmatrix} \sigma_t^2 & \frac{\alpha_t}{\alpha_{t'}} \sigma_{t'}^2 \\ \frac{\alpha_t}{\alpha_{t'}} \sigma_{t'}^2 & \sigma_{t'}^2 \end{pmatrix}\right)$$

Conversely, for the posterior flow case, we can derive the covariance terms conditioned on $X_1$ using (46) and the fact that $X_t, X_{s*}$ are independent given $X_1$:

$$\text{Cov}(X_t, X_{t'}|X_1) = \alpha_{t'} \frac{\bar{\alpha}_{s*} \sigma_t^2 \text{Cov}(X_{s*}, X_t|X_1) + \alpha_t \bar{\sigma}_{s*}^2 \text{Cov}(X_t, X_t|X_1)}{\bar{\alpha}_{s*}^2 \sigma_t^2 + \alpha_t^2 \bar{\sigma}_{s*}^2}$$

$$= \alpha_{t'} \frac{\alpha_t \bar{\sigma}_{s*}^2 \text{Cov}(X_t, X_t|X_1)}{\bar{\alpha}_{s*}^2 \sigma_t^2 + \alpha_t^2 \bar{\sigma}_{s*}^2}$$

$$= \alpha_{t'} \alpha_t \frac{\bar{\sigma}_{s*}^2 \text{Var}(X_t|X_1)}{\bar{\alpha}_{s*}^2 \sigma_t^2 + \alpha_t^2 \bar{\sigma}_{s*}^2}$$

$$= \alpha_{t'} \alpha_t \frac{\bar{\sigma}_{s*}^2 \sigma_t^2}{\bar{\alpha}_{s*}^2 \sigma_t^2 + \alpha_t^2 \bar{\sigma}_{s*}^2}$$

Finally, by construction

$$\sigma_{t'}^2 = \alpha_{t'}^2 g(t') = \alpha_{t'}^2 g(t^*(s^*, t)) = \alpha_{t'}^2 \frac{\sigma_t^2 \bar{\sigma}_{s*}^2}{\bar{\alpha}_{s*}^2 \sigma_t^2 + \alpha_t^2 \bar{\sigma}_{s*}^2} = \frac{\alpha_{t'}}{\alpha_t} \alpha_{t'} \alpha_t \frac{\bar{\sigma}_{s*}^2 \sigma_t^2}{\bar{\alpha}_{s*}^2 \sigma_t^2 + \alpha_t^2 \bar{\sigma}_{s*}^2} = \frac{\alpha_{t'}}{\alpha_t} \text{Cov}(X_t, X_t'|X_1)$$

Similarly, it holds that

$$\text{Var}(X_{t'}|X_1) = \alpha_{t'}^2 \frac{\bar{\alpha}_{s*}^2 \sigma_t^4 \bar{\sigma}_{s*}^2 + \alpha_t^2 \bar{\sigma}_{s*}^4 \sigma_t^2}{(\bar{\alpha}_{s*}^2 \sigma_t^2 + \alpha_t^2 \bar{\sigma}_{s*}^2)^2} = \alpha_{t'}^2 \frac{\bar{\alpha}_{s*}^2 \sigma_t^2 + \alpha_t^2 \bar{\sigma}_{s*}^2}{\bar{\alpha}_{s*}^2 \sigma_t^2 + \alpha_t^2 \bar{\sigma}_{s*}^2} \frac{\sigma_t^2 \bar{\sigma}_{s*}^2}{\bar{\alpha}_{s*}^2 \sigma_t^2 + \alpha_t^2 \bar{\sigma}_{s*}^2} = \alpha_{t'}^2 \frac{\sigma_t^2 \bar{\sigma}_{s*}^2}{\bar{\alpha}_{s*}^2 \sigma_t^2 + \alpha_t^2 \bar{\sigma}_{s*}^2} = \alpha_{t'}^2 g(t') = \sigma_{t'}^2$$

Therefore, it holds that

$$(X_t, X_{t'})|X_1 = x_1 \sim \mathcal{N}\left(\begin{pmatrix} \alpha_t x_1 \\ \alpha_{t'} x_1 \end{pmatrix}, \begin{pmatrix} \sigma_t^2 & \frac{\alpha_t}{\alpha_{t'}} \sigma_{t'}^2 \\ \frac{\alpha_t}{\alpha_{t'}} \sigma_{t'}^2 & \sigma_{t'}^2 \end{pmatrix}\right)$$

which coincides with distribution of the memoryless SDE conditioned on $X_1$. Therefore, they also must coincide if we make $X_1$ random with $p_{\text{data}}$. This proves that the joints coincide (see (42)). And therefore, also the distributions conditioned on $X_t = x_t$. This finishes the proof. $\square$

### A.9. Proof of Theorem 5.1

*Proof.* Let $X_{t,t'}$ be a standard (i.e. deterministic) ground-truth flow map. Recall the definitions

$$q_{t'|t}(x_{t'}|x_t) = \mathcal{N}\left(\frac{\alpha_{t'}}{\alpha_t} x_t; (\sigma_{t'}^2 - \frac{\alpha_{t'}^2}{\alpha_t^2} \sigma_t^2)I\right)$$

$$x_{t'}(x_t, \epsilon) = \frac{\alpha_{t'}}{\alpha_t} x_t + \left(\sqrt{\sigma_{t'}^2 - \frac{\alpha_{t'}^2}{\alpha_t^2} \sigma_t^2}\right) \epsilon.$$

By the construction of the flow map, if $x_{t'} \sim p_{t'}$, then it also holds that $z = X_{t',1}(x_{t'}) \sim p_{\text{data}}$. Hence, we can derive:

$$
\begin{aligned}
V_t(x_t) &= \log \int \exp(r(z)) \frac{p_t(x_t|z)p_{\text{data}}(z)}{p_t(x_t)} \mathrm{d}z \\
&= \log \int \exp(r(z))p_t(x_t|z)p_{\text{data}}(z)\mathrm{d}z - \log p_t(x) \\
&= \log \mathbb{E}_{z \sim p_{\text{data}}} \left[ \exp(r(z))p_t(x_t|z) \right] - \log p_t(x_t) \\
&= \log \mathbb{E}_{x_{t'} \sim p_{t'}} \left[ \exp(r(X_{t',1}(x_{t'})))p_t(x_t|X_{t',1}(x_{t'})) \right] - \log p_t(x_t) \\
&= \log \mathbb{E}_{x_{t'} \sim q_{t'|t}(\cdot|x_t)} \left[ \exp(r(X_{t',1}(x_{t'}))) \frac{p_t(x_t|X_{t',1}(x_{t'}))p_{t'}(x_{t'})}{q_{t'|t}(x_{t'}|x_t)} \right] - \log p_t(x_t) \\
&= \log \mathbb{E}_{x_{t'} \sim q_{t'|t}(\cdot|x_t)} \left[ \exp \Big( \underbrace{r(X_{t',1}(x_{t'}) + \log p_t(x_t|X_{t',1}(x_{t'})) + \log p_{t'}(x_{t'}) - \log q_{t'|t}(x_{t'}|x_t)}_{=: \tilde{v}_{t,t'}(x_t, x_{t'})} \Big) \right] - \log p_t(x_t) \\
&= \log \mathbb{E}_{\epsilon \sim \mathcal{N}(0,I_d)} \left[ \exp\left( \tilde{v}_{t,t'}(x_t, \epsilon) \right) \right] - \log p_t(x_t)
\end{aligned}
$$

By taking the gradient with respect to $x_t$, we obtain

$$
\begin{aligned}
\nabla_{x_t} V_t(x_t) &= \nabla_{x_t} \log \mathbb{E}_{\epsilon \sim \mathcal{N}(0,I_d)} \left[ \exp\left( \tilde{v}_{t,t'}(x_t, \epsilon) \right) \right] - \nabla_{x_t} \log p_t(x_t) \\
&= \frac{\mathbb{E}_{\epsilon \sim \mathcal{N}(0,I_d)} \left[ \exp\left( \tilde{v}_{t,t'}(x_t, \epsilon) \right) \left( \nabla_{x_t} \tilde{v}_{t,t'}(x_t, \epsilon) - \nabla_{x_t} \log p_t(x_t) \right) \right]}{\mathbb{E}_{\epsilon \sim \mathcal{N}(0,I_d)} \left[ \exp\left( \tilde{v}_{t,t'}(x_t, \epsilon) \right) \right]}
\end{aligned}
$$

As the score function $\nabla \log p_t(x_t)$ is tractable, it remains to show to obtain $\tilde{v}_{t,t'}(x_t, \epsilon)$ and $\nabla_{x_t} \tilde{v}_{t,t'}(x_t, \epsilon)$.

The gradient can be computed directly by taking the sum of the gradient of each individual term:

$$
\begin{aligned}
&\nabla_{x_t} \tilde{v}_{t,t'}(x_t, \epsilon) - \nabla_{x_t} \log p_t(x_t) \\
&= \nabla_{x_t}(r(X_{t',1}(x_{t'}(x_t, \epsilon_i))) + \log p_t(x_t|X_{t',1}(x_{t'}(x_t, \epsilon_i)))) + \nabla_{x_t} \log p_{t'}(x_{t'}(x_t, \epsilon)) - \nabla_{x_t} \log q_{t'|t}(x_{t'}(x_t, \epsilon_i)|x_t) - \nabla_{x_t} \log p_t(x_t) \\
&\overset{(i)}{=} \nabla_{x_t}(r(X_{t',1}(x_{t'}(x_t, \epsilon_i))) + \log p_t(x_t|X_{t',1}(x_{t'}(x_t, \epsilon_i)))) + \nabla_{x_{t'}} \log p_{t'}(x_{t'}) \frac{\alpha_{t'}}{\alpha_t} - 0 - \nabla_{x_t} \log p_t(x_t) \\
&= \nabla_{x_t} r_{\text{recov}}(x_t, \epsilon) + \delta_{\text{score}}(x_t, \epsilon)
\end{aligned}
$$

where in $(i)$ we used that $\log q_{t'|t}(x_{t'}(x_t, \epsilon)|x_t) = C - \frac{1}{2}\|\epsilon\|^2$ for a constant $C$ independent of $x_t$.

It remains to show how to obtain $\tilde{v}_{t,t'}(x_t, \epsilon)$. First, note that $\tilde{v}_{t,t'}(x_t, \epsilon)$ in itself is intractable as we do not know $p_{t'}(x_{t'})$. The key insight here is that we do not need to know the exact value of $\tilde{v}_{t,t'}(x_t, \epsilon)$ but only up to a constant in $\epsilon$. Specifically, we can compute:

$$
\begin{aligned}
\log p_{t'}(x_t') &= \log p_{t'}(x_t) + \int_0^1 \frac{\mathrm{d}}{\mathrm{d}u} \log p_{t'}(x_t + u(x_{t'} - x_t))\mathrm{d}u \\
&= \log p_{t'}(x_t) + \left[ \int_0^1 \nabla \log p_{t'}(x_t + u(x_{t'} - x_t))^T \mathrm{d}u \right]^T (x_{t'} - x_t) \\
&=: \log p_{t'}(x_t) + \gamma_{t,t'}(x_t, x_{t'}),
\end{aligned}
\tag{47}
$$

where above, $\gamma_{t,t'}(x_t, x_{t'})$ is the second term on the second line. Hence, we can compute that

$$
\tilde{v}_{t,t'}(x_t, \epsilon) = \underbrace{r_{\text{recov}}(x_t, \epsilon) + \log p_{t'}(x_t) + \frac{1}{2}\|\epsilon\|^2 + \gamma_{t,t'}(x_t, x_{t'})}_{=: v_{t,t'}(x_t, \epsilon)} + \log p_{t'}(x_t) + C = v_{t,t'}(x_t, \epsilon) + \log p_{t'}(x_t) + C
$$

While the green part is intractable, it can be dropped as it is a constant in $\epsilon$. Specifically, we obtain

$$\nabla_{x_t} V_t(x_t) = \frac{\mathbb{E}_{\epsilon \sim \mathcal{N}(0, I_d)} \left[ \exp\left(\tilde{v}_{t,t'}(x_t, \epsilon)\right) \left(\nabla_{x_t} \tilde{v}_{t,t'}(x_t, \epsilon) - \nabla_{x_t} \log p_t(x_t)\right) \right]}{\mathbb{E}_{\epsilon \sim \mathcal{N}(0, I_d)} \left[ \exp\left(\tilde{v}_{t,t'}(x_t, \epsilon)\right) \right]} \tag{48}$$

$$= \frac{\exp(\log p_{t'}(x_t) + C)}{\exp(\log p_{t'}(x_t) + C)} \frac{\mathbb{E}_{\epsilon \sim \mathcal{N}(0, I_d)} \left[ \exp\left(v_{t,t'}(x_t, \epsilon)\right) \left(\nabla_{x_t} \tilde{v}_{t,t'}(x_t, \epsilon) - \nabla_{x_t} \log p_t(x_t)\right) \right]}{\mathbb{E}_{\epsilon \sim \mathcal{N}(0, I_d)} \left[ \exp\left(v_{t,t'}(x_t, \epsilon)\right) \right]} \tag{49}$$

$$= \frac{\mathbb{E}_{\epsilon \sim \mathcal{N}(0, I_d)} \left[ \exp\left(v_{t,t'}(x_t, \epsilon)\right) \left(\nabla_{x_t} \tilde{v}_{t,t'}(x_t, \epsilon) - \nabla_{x_t} \log p_t(x_t)\right) \right]}{\mathbb{E}_{\epsilon \sim \mathcal{N}(0, I_d)} \left[ \exp\left(v_{t,t'}(x_t, \epsilon)\right) \right]} \tag{50}$$

By taking Monte Carlo samples, we obtain the theorem. $\qquad\square$

## A.10. Value function estimation with Weighted Diamond Map

Finally, we remark that one could also estimate the value function (instead of its gradient) via a weighted Diamond map. By the Continuity equation

$$\log p_t(x_t) - \log p_{t'}(x_t)$$

$$= \int_{t'}^{t} \partial_r \log p_r(x_t) \mathrm{d}r$$

$$= \int_{t'}^{t} -\nabla \log p_r(x_t)^T u_r(x_t) - \nabla \cdot u_r(x_t) \mathrm{d}r$$

$$= -\mathbb{E}_{r \sim \mathrm{Unif}_{[t',t]}, \epsilon \sim \mathcal{N}(0, I_d)} \left[ \nabla \log p_r(x_t)^T u_r(x_t) + \epsilon^T D_{x_t} u_r(x_t) \epsilon \right]$$

Therefore, we know that:

$$V_t(x_t)$$

$$= \log \mathbb{E}_{q_{t'|t}(\cdot|x_t)} \left[ \exp(r(z) + \gamma_t(x_{t'})) \frac{p_t(x_t|z)}{q_{t'|t}(x_{t'}|x_t)} \right] + \mathbb{E}_{r \sim \mathrm{Unif}_{[t',t]}, \epsilon \sim \mathcal{N}(0, I_d)} \left[ \nabla \log p_r(x_t)^T u_r(x_t) + \epsilon^T D_{x_t} u_r(x_t) \epsilon \right]$$

The last term requires backpropagation and constitutes a Hutchinson trace estimator term. Hence, this might be less efficient, yet note that the Hutchinson's trace estimator is for fixed $x_t$ (does not depend on the sample $x_{t'}$ that is drawn) and therefore the second summand constitutes a fixed offset.

# B. Details on Diamond Maps

## B.1. Search or Sequential Monte Carlo (SMC) with Posterior Diamond Maps

We discuss Sequential Monte Carlo (SMC) with Diamond Maps. SMC evolves $M$ particles $X_t^1, \cdots, X_t^M$ simultaneously. To evolve them in time, we use a proposal distribution given by the DDPM transitions either sampled via Theorem 4.3 in one step. Further, resampling happens based on potentials given by $U_{t,t'}(x_t, x_{t'})$. Here, we choose $U_{t,t'}(x_t, x_{t'}) = V_{t'}^r(x_{t'}) - V_t(x_t)$, i.e. the potentials favors particles promising to increase the value function most. We can estimate the value function $V_t(x_t)$ via

$$V_t(x_t) \approx \log \frac{1}{N} \sum_{i=1}^{N} \exp(r(z^i)) \quad z^i = X_{0,1}^\theta(\bar{x}_0^i | x_t, t) \tag{51}$$

where $\bar{x}_0^i \sim \mathcal{N}(0, I_d)$ for every $i = 1, \cdots, N$. The full method is summarized in Algorithm 2.

Same as for guidance, achieving the same estimators of the value function would be $\mathcal{O}(N)$ steps more expensive for a standard flow model. Here, this speed-up is explained by two factors: (1) Transitions $p_{t'|t}^{\mathrm{DDPM}}$ can be obtained in one step (2) Samples $p_{1|t}$ can be obtained in one step.

**Remark - Search.** We can apply Diamond Maps similarly to search and obtain the same speed-up. Search proceeds similar to SMC but unlike SMC it does not keep a population of particles but collapses all particles into a single particle with

**Algorithm 2** Sequential Monte Carlo with Diamond Maps

1: **Require:** Diamond flow $X_{s,s'}^\theta(\bar{x}_s|x_t, t)$, $N$ transitions, $M$ num. of partic., Monte Carlo samples $K$, reward $r$
2: **Init:** $x_0^m \sim \mathcal{N}(0, I_d), V_0^m = U_0^m = 0 \quad (m \leq M)$
3: **Set** $h \leftarrow 1/N, t \leftarrow 0$
4: **for** $n = 0$ to $N - 1$ **do**
5:     **for** $m = 1$ to $M$ **do**
6:         $x_{t+h}^m \sim p_{t+h|t}^{\mathrm{DDPM}}(\cdot|x_t^m)$ (via Theorem 4.3)
7:         $V_{t+h}^m \leftarrow 0$
8:         **for** $k = 1$ to $K$ **do**
9:             $z^k \leftarrow X_{0,1}^\theta(\bar{x}_0^{k,m}|x_{t+h}, t+h) \quad (\bar{x}_0^{k,m} \sim \mathcal{N}(0, I_d))$
10:            $V_{t+h}^m \leftarrow V_{t+h}^m + \frac{1}{K}\exp(r(z^k))$
11:         **end for**
12:         $V_{t+h}^m = \log V_{t+h}^m$
13:         $U_{t+h}^m \leftarrow U_t^m + V_{t+h}^m - V_t^m$
14:     **end for**
15:     $t \leftarrow t + h$
16:     **if** $\mathrm{ESS}(U_t^1, \cdots, U_t^M) <$ threshold **then**
17:         **Resample particles:** $i_k \sim \mathrm{softmax}(U_t^1, \ldots, U_t^M)$
18:         $U_t^i \leftarrow 0$
19:         $V_t^k = V_t^{i_k}$
20:         $x_t^k \leftarrow x_t^{i_k}, V_t^k \leftarrow V_t^{i_k}$
21:     **end if**
22: **end for**
23: **Return:** $x_1^1, \cdots, x_1^M$

maximum potential, i.e. the softmax is taken with respect to zero temperature. With this difference, Algorithm 2 can be easily modified to apply Diamond Maps to search methods.

### B.2. General GLASS transitions

**General transitions.** We note that our framework would also work for general GLASS transitions (i.e. not limited to the DDPM transitions) and we could similarly distill these transitions into a flow map. However, we would need to condition the flow map on an additional input $t'$, which adds complexity and redundancy to the learning problem that we choose to avoid here by leveraging Theorem 4.3.

### B.3. Guidance with Weighted Diamond Maps

We present an algorithm in Algorithm 3.

## C. Extended Related Work

**Flow Maps.** Many recent works have explored training methods of flow maps and consistency models (Boffi et al., 2025a; Geng et al., 2025; Song et al., 2023c; Song & Dhariwal, 2023). While training methods may differ, all of these approaches rely on distillation of the same ODE trajectory of the marginal vector field, also called probability flow ODE in the diffusion literature. Here, we show how one can construct a *stochastic* flow map, i.e. one that it not fully determined by the current state $x_t$.

**Distilled Models for reward alignment.** The LATINO sampler (Spagnoletti et al., 2025) iteratively noises a samples $z$ and then denoises them with a one-step sampler.

**Inference-time reward alignment.** Guidance methods (Chung et al., 2022; Abdolmaleki et al.; Ye et al., 2024; Yu et al., 2023; Bansal et al., 2023; He et al., 2023; Graikos et al., 2022; Song et al., 2023b;a; Feng et al., 2025) were used particularly for solving inverse problems such as Gaussian deblurring or inpainting. The inaccurate denoiser approximation (see (10)) is

| | CIFAR-10 | CelebA-64 | | ImageNet1k |
|---|---|---|---|---|
| *Dataset Properties* | | | *Dataset Properties* | |
| Dimensionality | $3 \times 32 \times 32$ | $3 \times 64 \times 64$ | Dimensionality | $3 \times 256 \times 256$ |
| Samples | 50k | 203k | Samples | 1281167 |
| | | | Latent Size | $4 \times 32 \times 32$ |
| *Network* | | | Encoder | sd-vae-ft-ema |
| Architecture | EDM2 | EDM2 | | |
| Hidden/base channels | 128 | 128 | *Network* | |
| Channel multipliers | [2, 2, 2] | [1, 2, 3, 4] | Architecture | SiT B/2 |
| Residual blocks | 4 per resolution | 3 per resolution | CFG | 4 |
| Attention resolutions | $16 \times 16$ | $16 \times 16, 8 \times 8$ | | |
| Dropout | 0.13 | 0.0 | *Hyperparameters* | |
| | | | Batch size | 256 |
| *Hyperparameters* | | | Training steps | 320,000 |
| Batch size | 512 | 256 | Optimizer | RAdam |
| Training steps | 400,000 | 800,000 | Learning rate | $10^{-4}$ |
| Optimizer | RAdam | RAdam | LR schedule | Constant |
| Learning rate | $10^{-2}$ | $10^{-2}$ | Gradient clipping | 1.0 |
| LR schedule | Sqrt decay at 35k | Sqrt decay at 35k | Diagonal fraction | 0.75 |
| Gradient clipping | 1.0 | 1.0 | EMA decay | 0.9999 |
| Diagonal fraction | 0.75 | 0.75 | | |
| EMA decay | 0.9999 | 0.9999 | *Evaluation* | |
| | | | Metric | FID |
| *Evaluation* | | | Sample count | 50,000 |
| Metric | FID | FID | | |
| Sample count | 50,000 | 50,000 | *Methods* | |
| | | | Loss | LSD |
| *Methods* | | | Initialized From | MeanFlow B/2 w/ CFG 2 |
| Loss | LSD | LSD | Teacher Model | Flow SiT L/2 |
| | | | | |
| *Compute* | | | *Compute* | |
| GPU | 8x L40S | 6x 40GB A100 | GPU | 8x L40S |
| Precision | fp32 | bfloat16 | Precision | bfloat16 |

*Table 1.* **Experimental setup**. Flow map (for CelebA64) and Posterior Diamond Map both trained using these configurations. Posterior Diamond Map contains additional embeddings and channels for extra time steps and conditioning variables as described in (40).

a known limitation of guidance methods and many methods aim to find better - yet biased - approximations (Song et al., 2023a). Here, we are taking a more "radical" step of learning the posterior directly. SMC (Singhal et al., 2025; Skreta et al., 2025; Wu et al., 2023; Mark et al., 2025; He et al., 2025) and search (Li et al., 2025b; Zhang et al., 2025) evolve a population of particles and are based on filtering out unpromising particles. Their main challenges are expensive or biased estimation of the value function $V_t(x_t)$ or a collapse of diversity of the population of particles. Many other approaches (Yeh et al., 2024; Wu et al., 2024; Krishnamoorthy et al., 2023) exist and we refer to (Uehara et al., 2025) for a review.

**Reward fine-tuning.** Reward fine-tuning methods fine-tune flow and diffusion models at training based on a variery of RL techniques such as GRPO (Xue et al., 2025; Li et al., 2025a; Liu et al.), stochastic optimal control (Liu et al.; Domingo-Enrich et al., 2024), DPO (Wallace et al., 2024). Many of these methods require expensive simulation and stochastic transitions for exploration during training (Liu et al., 2025; Xue et al., 2025; Li et al., 2025a; Domingo-Enrich et al., 2024) that we learn in a one-step sampler.

# D. Experimental Details and Further Experiments

Code for our experiments can be found under `https://github.com/PeterHolderrieth/diamond_maps`.

### D.1. Posterior Diamond Map

#### D.1.1. CIFAR10/CELEBA-64

**Architecture** There are 2 relevant networks for CIFAR10 and CelebA-64, both unconditional and based on the EDM2 architecture (Karras et al., 2023): a teacher network and the Posterior Diamond Map. The teacher network is a flow map using the EDM2 architecture with configuration from Table 1 that has time parameters $t, t'$ and input $x_t$. The Posterior Diamond Map is a similar EDM2 architecture but with time parameters $t, t', s, s'$, input $\bar{x}_s$, and conditioning variable $x_t$.

**Experimental Setup** We fix $t' = 1$ for easier training but can still sample arbitrary $t'$ using Diamond DDPM Early stop sampling. Because of singularities for calculating the time reparameterization (18) and the GLASS velocity field (14), we limit $t$ to $[0.01, 0.99]$ during training and sampling. We sample $t$ uniformly from $U([0.01, 0.99])$ and $s, s'$ uniformly from the upper triangle of $[0.01, 0.99]^2$. We embed $t, t' - t$ and $s, s' - s$ based on empirical benefits seen in (Boffi et al., 2025b). We also similarly employ loss reweighting but calculate weights with all 4 diamond time steps instead of just 2 for a flow map. We use linear outer time schedulers $\alpha_t = t, \sigma_t = 1 - t$ and the corresponding inner time schedulers $\bar{\alpha}_s, \bar{\sigma}_s$ from (Holderrieth et al., 2025). We compute FID for 4 different NFEs: $1, 2, 4, 8$ (Table 2). The flow method uses the flow map but gets the velocity by purely using the diagonal $t = t'$. The flow map method uses the vanilla flow map sampling with $t \to t'$ where $t < t'$. GLASS follows the same sampling procedure as (Holderrieth et al., 2025) and uses the same velocity as the flow. Lastly, we sample from the $p_{1|0}^{\text{DDPM}}(x_1|x_0)$ transition kernel with inner steps equal to NFE for the Posterior Diamond Map.

For the posterior samples, we compare the unconditional EDM2 Posterior Diamond Map to the unconditional EDM2 flow map.

| Dataset | Method | Step Count | | | |
|---------|--------|:------:|:------:|:------:|:------:|
| | | 1 | 2 | 4 | 8 |
| CIFAR-10 (FID ↓) | Flow | 402.049 | 145.275 | 41.231 | 14.187 |
| | Flow Map | **10.604** | **4.596** | **4.180** | **4.880** |
| | GLASS | 378.674 | 157.554 | 39.472 | 11.597 |
| | Diamond Map | 12.551 | 5.799 | 5.803 | 6.732 |
| CelebA-64 (FID ↓) | Flow | 199.933 | 89.582 | 45.461 | 24.762 |
| | Flow Map | **9.534** | **4.052** | **3.083** | **3.146** |
| | GLASS | 208.509 | 95.245 | 51.803 | 26.333 |
| | Diamond Map | 16.084 | 9.160 | 6.736 | 5.776 |
| ImageNet (FID ↓) | Flow | 266.22 | 135.85 | 16.90 | 11.81 |
| | Flow Map | **6.06** | **5.22** | **5.36** | **5.51** |
| | Diamond Map | 10.74 | 9.67 | 10.10 | 10.44 |

*Table 2.* **Pretraining results**. FID across various step counts for CIFAR-10, CelebA-64, and ImageNet datasets. Best method per dataset and step count shown in **bold**. As one can see, the flow map contained in a Posterior Diamond Map performs almost on par with the standard flow Map.

#### D.1.2. IMAGENET

**Architecture** There are 3 relevant networks for ImageNet, all conditional networks based on SiT (Ma et al., 2024): a teacher network, an initialization network, and the Posterior Diamond Map. The teacher network is a pretrained flow-matching network using the vanilla SiT architecture with time parameters $t$ and input $x_t$. It uses CFG at inference time, unlike the following models, and it's scale is decided by the target CFG scale for the Posterior Diamond Map. The initialization network is a pretrained MeanFlow flow map based on the SiT architecture with time parameters $t, t'$ and input $x_t$. The Posterior Diamond Map is a modified SiT architecture with time parameters $t, t', s, s'$, input $\bar{x}_s$, and conditioning variable $x_t$. Network parameters for all 3 networks can be found in Table 1.

**Experimental Setup** We use the same setup for time step sampling, schedules, and metric sampling as Section D.1.1. For training the Posterior Diamond Map, we first initialize it off a pretrained flow map, then distill the GLASS velocity field equation 14 from a teacher. We initialize a Posterior Diamond Map from a pretrained flow map by adding 2 additional time

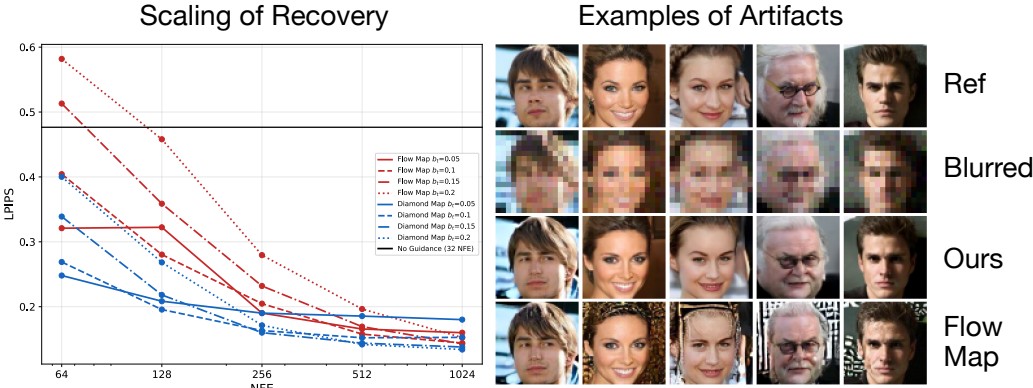

*Figure 8.* Applying Posterior Diamond Maps (see Section 4.1) to super resolution for various reward scales $b_t$. Posterior Diamond Maps has a better Pareto frontier of optimal results for optimal compute. The reason for that is that we find is that Posterior Diamond Maps shows significantly higher robustness towards high reward scales, likely caused by stochasticity, compared to guidance with a naive flow map approximation.

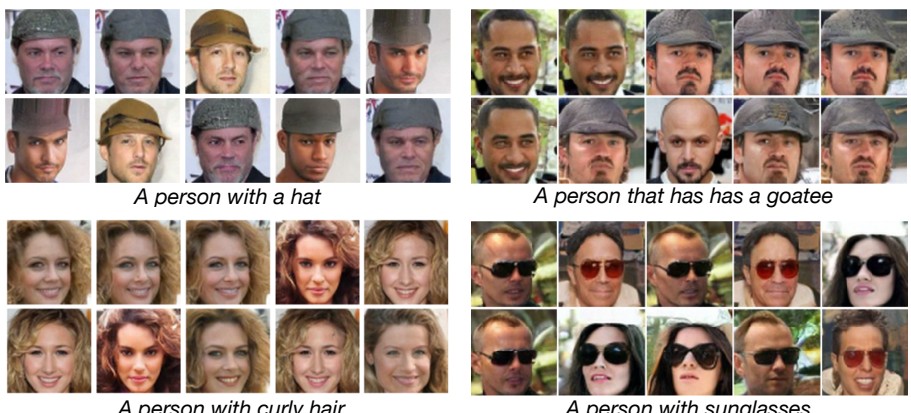

*Figure 9.* Qualitative examples of CLIPScore reward alignment with Sequential Monte Carlo using Posterior Diamond Maps for CelebA64.

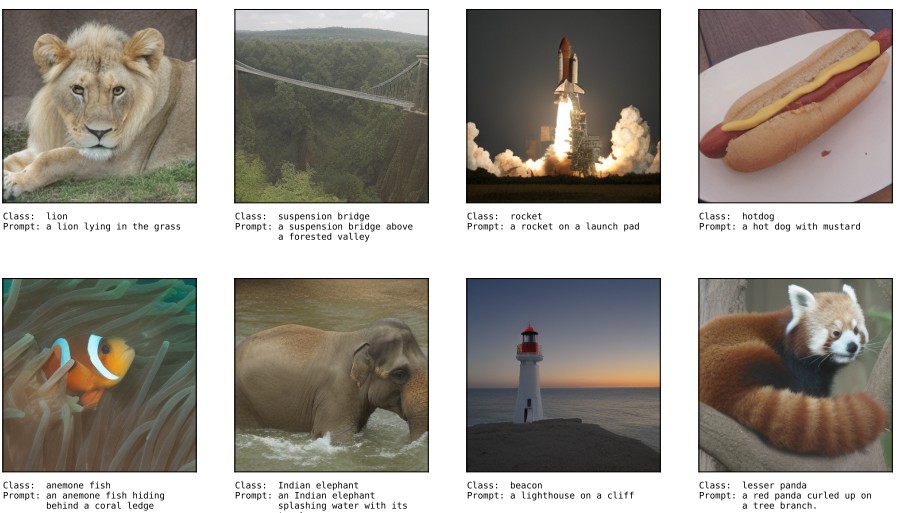

*Figure 10.* Qualitative examples of ImageReward-reward alignment with Sequential Monte Carlo with Posterior Diamond Maps. Model trained on ImageNet1k. We extend class-conditioning to conditioning with simple prompts.

| Dataset | Time $t$ | Steps $N$ | | | | | | | |
| | | $N=1$ | | $N=2$ | | $N=4$ | | $N=8$ | |
| | | GLASS | Diamond | GLASS | Diamond | GLASS | Diamond | GLASS | Diamond |
|---|---|---|---|---|---|---|---|---|---|
| CelebA-64 (FID ↓) | $t=0.1$ | 283.278 | **11.731** | 97.932 | **5.624** | 21.843 | **5.773** | 8.796 | **6.876** |
| | $t=0.2$ | 118.422 | **9.719** | 46.968 | **5.000** | 15.371 | **5.420** | 8.574 | **6.360** |
| | $t=0.3$ | 57.299 | **7.104** | 29.438 | **3.942** | 13.361 | **4.245** | 7.928 | **4.921** |
| | $t=0.5$ | 23.727 | **4.359** | 16.944 | **2.867** | 10.674 | **2.825** | 6.690 | **3.117** |
| ImageNet (FID ↓) | $t=0.1$ | 213.559 | **9.911** | 86.231 | **8.505** | 12.360 | **9.014** | 9.587 | **9.227** |
| | $t=0.2$ | 130.868 | **8.941** | 48.107 | **6.230** | 10.225 | **6.427** | 7.334 | **6.492** |
| | $t=0.3$ | 79.131 | **7.991** | 28.185 | **4.678** | 8.385 | **4.420** | 5.839 | **4.469** |
| | $t=0.5$ | 18.660 | **4.619** | 9.161 | **2.797** | 4.807 | **2.320** | 3.380 | **2.257** |

*Table 3.* **Posterior recovery results**. FID across step counts for GLASS and Posterior Diamond Map at different timesteps $t$. Bold indicates the better method at each (dataset, $t$, $N$) combination.

| Dataset | Method | Step Count | | | | | |
| | | 1 | 2 | 3 | 5 | 10 | 20 |
|---|---|---|---|---|---|---|---|
| CelebA-64 (FID ↓) | Renoising | 16.316 | 11.107 | 10.239 | 10.593 | 12.728 | 17.071 |
| | Early Stopping | **16.292** | **10.256** | **8.614** | **7.699** | **7.763** | **8.444** |
| ImageNet (FID ↓) | Renoising | 10.783 | 10.186 | 10.793 | 11.799 | 13.212 | 14.385 |
| | Early Stopping | **10.739** | **9.878** | **10.447** | **11.167** | **11.784** | **12.007** |

*Table 4.* FID Comparison between renoising and early stopping for CelebA-64 and ImageNet datasets. Best method at each step count is shown in **bold**.

embeddings for $s, s'$ and an additional image embedding for $\bar{x}_s$. These embeddings are then mixed with the original time embeddings $(t, t')$ and image embeddings $(x_t)$ with separate residual MLPs. The training steps in Table 1 refer to extra training steps and does not include the training steps associated with the initialization network. We use conditional networks for ImageNet, which allows us to use CFG (Ho & Salimans, 2022). The Posterior Diamond Map formulation allows us to naturally distill from a GLASS velocity field with CFG, giving us 1-NFE CFG sampling. We use the sd-vae-ft-ema encoder for all ImageNet networks. We refer to (Ma et al., 2024) and (Geng et al., 2025) for training details of the teacher and initialization networks.

For the posterior samples, we compare the B/2 Posterior Diamond Map with CFG 4 to a flow SiT B/2 with CFG (at inference time) 4 and a flow map B/2 with CFG 2. We choose to use the B/2 model size because of limited compute resources and well trained checkpoints for the B/2 model size.

### D.1.3. GUIDANCE

For CelebA-64, we use the diagonal of the flow map for our base drift. For ImageNet, we use a flow SiT XL/2 network with CFG 4 (at inference time). There are 2 main parameters for guidance: the guidance scale $b_t$ and the number of samples to use for estimation $K_t$ at each timestep. For CelebA-64, we ablate over a constant $b_t \in \{0.05, 0.1, 0.15, 0.2\}$. For ImageNet, we normalize the guidance term to the norm of the base drift up to a parameter $b_t$:

$$u_t^r = u_t(x) + b_t \cdot \frac{\|u_t(x)\|}{\|\nabla_{x_t} V_t(x_t)\|} \cdot \nabla_{x_t} V_t(x_t)$$

We find in practice that this stabilizes the guidance process. For $K_t$, we use a constant 1 Monte Carlo (MC) posterior samples for flow, flow map. For Diamond posterior sampling, we ablate over various MC sample schedules, such as a front loaded schedule with a majority of MC samples concentrated at the beginning of guidance. We keep MC samples greater than 0 at every time step for all ablations. For metric results, we take the best result over the ablated parameters.

**NFE** Given $K$ base drift steps, $N_k$ MC samples at each step, and $M$, the posterior sample cost, we have the following NFE

| LPIPS ↓ | | | | | | | | |
|---|---|---|---|---|---|---|---|---|
| Method | 12 | 36 | 48 | 72 | 144 | 288 | 504 | 768 | 1008 |
| DPS | 0.768 | 0.658 | 0.640 | 0.625 | **0.614** | 0.614 | 0.622 | 0.624 | 0.629 |
| FMAP | **0.716** | **0.645** | **0.636** | **0.622** | 0.618 | 0.620 | 0.628 | 0.637 | 0.638 |
| DIAMOND ($N_k = 1$) | 0.747 | 0.648 | 0.640 | 0.625 | 0.616 | 0.616 | 0.614 | 0.616 | 0.617 |
| DIAMOND | 0.773 | 0.660 | 0.645 | 0.631 | 0.618 | **0.608** | **0.611** | **0.605** | **0.606** |

| KID ↓ | | | | | |
|---|---|---|---|---|---|
| Method | 12 | 36 | 48 | 72 | 144 |
| DPS | $7.26 \times 10^{-2}$ | $-1.54 \times 10^{-3}$ | $-1.34 \times 10^{-3}$ | $-1.62 \times 10^{-3}$ | $-1.46 \times 10^{-3}$ |
| FMAP | $3.41 \times 10^{-2}$ | $3.88 \times 10^{-4}$ | $-1.05 \times 10^{-3}$ | $-1.05 \times 10^{-3}$ | $-1.02 \times 10^{-3}$ |
| DIAMOND ($N_k = 1$) | $\mathbf{1.85 \times 10^{-2}}$ | $4.11 \times 10^{-4}$ | $9.92 \times 10^{-4}$ | $5.88 \times 10^{-4}$ | $8.72 \times 10^{-4}$ |
| DIAMOND | $5.14 \times 10^{-2}$ | $9.45 \times 10^{-4}$ | $1.41 \times 10^{-3}$ | $5.55 \times 10^{-4}$ | $3.45 \times 10^{-4}$ |

| KID ↓ | | | | |
|---|---|---|---|---|
| Method | 288 | 504 | 768 | 1008 |
| DPS | $\mathbf{-1.51 \times 10^{-3}}$ | $8.11 \times 10^{-4}$ | $2.13 \times 10^{-3}$ | $3.11 \times 10^{-3}$ |
| FMAP | $-1.12 \times 10^{-3}$ | $\mathbf{-7.59 \times 10^{-4}}$ | $3.61 \times 10^{-3}$ | $2.33 \times 10^{-3}$ |
| DIAMOND ($N_k = 1$) | $1.56 \times 10^{-3}$ | $1.33 \times 10^{-3}$ | $\mathbf{-1.48 \times 10^{-3}}$ | $\mathbf{-1.62 \times 10^{-3}}$ |
| DIAMOND | $1.95 \times 10^{-4}$ | $-1.55 \times 10^{-4}$ | $4.81 \times 10^{-4}$ | $-5.47 \times 10^{-4}$ |

*Table 5.* Super resolution 32x results across methods and NFE for LPIPS and KID on the ImageNet1k validation split. Best metric at each NFE is shown in bold. We use $2 \cdot K + \sum_k N_k M$ with $M = 2$ for DPS and $M = 1$ otherwise for the NFEs in this table. This is used to create a standard set of NFEs that all the results can map to exactly.

for a single sample:

$$\text{NFE}_{\text{guidance}} = \underbrace{(1 + \mathbb{1}_{\text{cfg}})K}_{\text{base cost}} + \underbrace{\sum_k 3MN_k}_{\text{MC cost}}$$

For CelebA-64, we have a single network size, so we establish a forward pass of the EDM2 network as 1 NFE. For ImageNet, we use different network sizes, and we establish a forward pass of the XL network (largest) as 1 NFE. We establish the NFE values of other networks as a linear scaling with respect to the number of parameters using the XL/2 network as the baseline. For example, a B/2 network is treated as $1/5$ NFE. The base cost is a single forward pass over the flow network (2 forward passes if using CFG). The MC cost is multiplied by 3 to account for a forward pass for the posterior sample and a backward pass to estimate the gradient (approximately 2 forward passes). $M = 2$ for DPS for inference time CFG and $M = 1$ otherwise.

**Inverse Problems** We evaluate guidance using Posterior Diamond Map on noisy linear inverse problems. We follow the same formula as (Chung et al., 2022). For CelebA-64, we use super resolution 4x with mean downsampling. We calculate metrics over 1000 samples. Refer to Figure 8 for the experimental results. For ImageNet, we use super resolution 32x with bicubic downsampling. Both settings use Gaussian noise with $0.05^2$ variance. We ablate over $b_t \in \{1, 2, 4, 8, 10\}$. We calculate metrics over 64 samples in the ImageNet1k validation set. Refer to Figure 12, Figure 13, and Table 5 for experimental results.

**Prompt Alignment** We evaluate guidance using Posterior Diamond Map on prompt alignment problems only for ImageNet. We use 2 rewards, ImageReward (Xu et al., 2023) and CLIPScore (Hessel et al., 2021), optimizing one and evaluating the other as a held out reward. For visualizations, we also use a composite reward (Eyring et al., 2024). We ablate over $b_t \in \{1, 1.5, 2, 3, 5, 8\}$. We calculate metrics over 30 prompts, generating 8 samples each for Figure 14. For qualitative examples, refer to Figure 19.

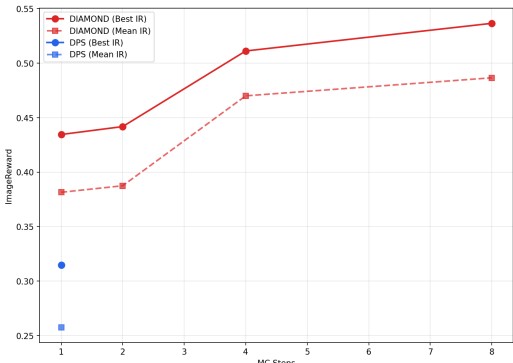

*Figure 11.* **Posterior Diamond Map scaling with Monte Carlo (MC) samples**. We scale MC samples for Sequential Monte Carlo on ImageNet1k. We fix an approximate NFE budget of 1000 and 6 outer steps and ablate batch size $B$ and the number of MC samples accordingly. We see that increasing MC samples improves performance even at the cost of a lower batch size. This demonstrates MC samples as a viable axis of scaling outside of batch size. The exact parameters in the format of (MC, B, NFE) are as follows, DIAMOND: $(1, 10, 970), (2, 9, 882), (4, 8, 800), (8, 7, 728)$; DPS: $(1, 10, 970)$.

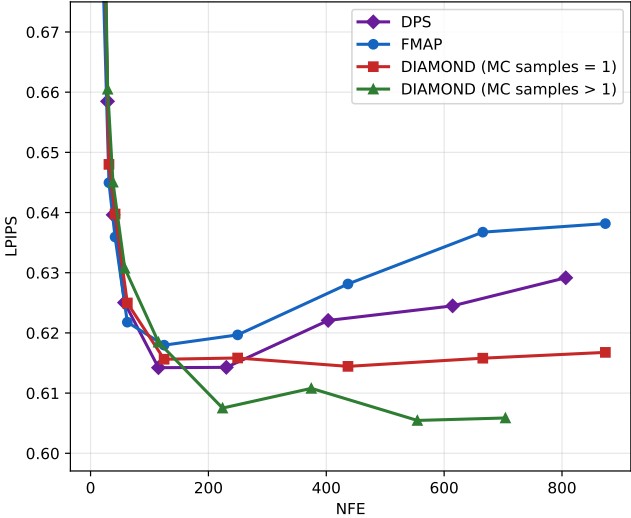

*Figure 12.* **Posterior Diamond Maps for Super Resolution on ImageNet1k.** Quantitative results for super resolution 32x using the setup at Section D.1.3. All results are chosen as the best out of ablations. For MC samples $> 1$, we ablate over various front loaded MC sample schedules. See qualitative examples are in Figure 13.

---

**Algorithm 3** Guidance with Weighted Diamond Maps

---

1: **Require:** Pre-trained flow map $X_{t,t'}(x_t)$ and flow model $u_t(x)$ (usually, flow map $X_{t,t'}(x)$ includes $u_t(x)$), $N$ simulation steps, $K$ Monte Carlo samples, differentiable reward $r$, guidance window $0 \leq t_{\text{guid-min}} \leq t_{\text{guid-max}} \leq 1$. $\lambda$ gradient norm scale.

2: **Init:** $x_0 \sim \mathcal{N}(0, I_d)$

3: **Set** $h \leftarrow 1/N, t \leftarrow 0$

4: **for** $n = 0$ to $N - 1$ **do**

5: $\quad t' = g^{-1}\left(\lambda \frac{\sigma_t^2}{\alpha_t^2}\right)$ (increase signal-to-noise ratio by a factor of $\lambda$)

6: $\quad v_t \leftarrow u_t(x)$

7: $\quad$ **if** $t \in [t_{\text{guid-min}}, t_{\text{guid-max}}]$ **then**

8: $\quad\quad v_{t',t} \leftarrow u_{t'}(x)$

9: $\quad\quad s_t \leftarrow \text{ConversionVelocityToScore}(v_t, t)$

10: $\quad\quad s_{t'} \leftarrow \text{ConversionVelocityToScore}(v_{t',t}, t')$

11: $\quad\quad$ **for** $k = 1$ to $K$ **do**

12: $\quad\quad\quad x_{t'}^k \leftarrow \frac{\alpha_{t'}}{\alpha_t} x_t + \sqrt{\sigma_{t'}^2 - \frac{\alpha_{t'}^2}{\alpha_t^2} \sigma_t^2} \epsilon^k \quad (\epsilon^k \sim \mathcal{N}(0, I_d))$

13: $\quad\quad\quad z^k \leftarrow X_{t',1}^\theta(x_{t'}^k)$

14: $\quad\quad\quad r_{\text{recov}}^k = r(z^k) - \frac{\|x_t - \alpha_t z^k\|^2}{2\sigma_t^2}$

15: $\quad\quad\quad \nabla_{x_t} r_{\text{recov}}^k \leftarrow r_{\text{recov}}^k.\text{backward}(x_t)$

16: $\quad\quad\quad$ **if** UseTweedieApproximation **then**

17: $\quad\quad\quad\quad s_{t'}^k = \frac{\alpha_{t'} z^k - x_{t'}}{\sigma_{t'}^2}$

18: $\quad\quad\quad$ **else**

19: $\quad\quad\quad\quad v_{t'}^k \leftarrow u_{t'}(x_{t'}^k)$

20: $\quad\quad\quad\quad s_{t'}^k \leftarrow \text{ConversionVelocityToScore}(v_{t'}^k, t')$

21: $\quad\quad\quad$ **end if**

22: $\quad\quad\quad \delta_{\text{score}}^i = s_{t'}^k \frac{\alpha_{t'}}{\alpha_t} - s_t$

23: $\quad\quad\quad \gamma_k \leftarrow \frac{1}{2} \left(s_{t'}^k + s_{t'}\right)^T (x_{t'}^k - x_t)$

24: $\quad\quad\quad v_k \leftarrow r(z^k)$ approximate with only reward weights

25: $\quad\quad$ **end for**

26: $\quad\quad g_{\text{reward}} \leftarrow \sum_k \text{softmax}(\ldots)[k] \nabla_{x_t} r(z^k)$

27: $\quad\quad g_{\text{likelihood}} \leftarrow \sum_k \text{softmax}(\ldots)[k] \nabla_{x_t} \left(-\frac{\|x_t - \alpha_t z^k\|^2}{2\sigma_t^2}\right)$

28: $\quad\quad g_{\text{score}} \leftarrow \sum_k \text{softmax}(\ldots)[k] \delta_{\text{score}}^k$

29: $\quad\quad$ **if** Normalize **then**

30: $\quad\quad\quad g_{\text{reward}} \leftarrow \frac{g_{\text{reward}}}{\|g_{\text{reward}}\|}$

31: $\quad\quad\quad g_{\text{likelihood}} \leftarrow \frac{g_{\text{likelihood}}}{\|g_{\text{likelihood}}\|}$

32: $\quad\quad\quad g_{\text{score}} \leftarrow \frac{g_{\text{score}}}{\|g_{\text{score}}\|}$

33: $\quad\quad$ **end if**

34: $\quad\quad \nabla_{x_t} V_t(x_t) \leftarrow \lambda \left(g_{\text{reward}} + g_{\text{likelihood}} + g_{\text{score}}\right)$

35: $\quad\quad u_t^r \leftarrow v_t + b_t \nabla_{x_t} V_t(x_t)$

36: $\quad$ **else**

37: $\quad\quad u_t^r = v_t$

38: $\quad$ **end if**

39: $\quad X_{t+h} \leftarrow X_t + h u_t^r$

40: $\quad t \leftarrow t + h$

41: **end for**

42: **Return:** $X_1$

---

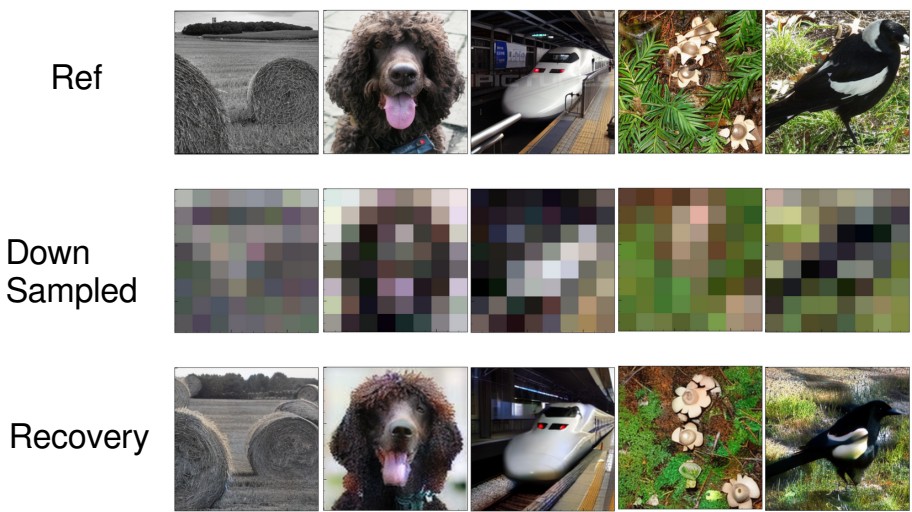

*Figure 13.* Qualitative examples of super resolution 32x using the Posterior Diamond Map with guidance on the ImageNet1k validation set.

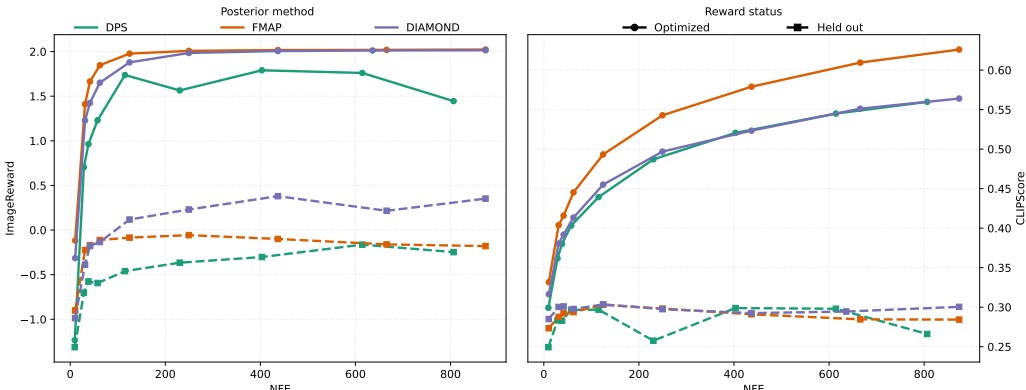

*Figure 14.* Applying Posterior Diamond Maps for prompt alignment (see Section D.1.3). We find that the Posterior Diamond Map is less prone to reward hacking (measured via a held out reward), while exhibiting similar performance for the optimized reward. We may expect the gap to come from weaker reward optimization from the higher FID samples from the Posterior Diamond Map. However, DPS has worse FID compared to the Posterior Diamond Map yet performs worse on the held-out reward compared to the flow map. This suggests the improvement comes from better posterior samples rather than simply weaker optimization.

### D.1.4. SMC

For CelebA-64, we use the Posterior Diamond Map as our base drift. For ImageNet, we use GLASS on top of the same flow SiT XL/2 network with CFG 4 (at inference time) as in Section D.1.3 as our base drift. Using GLASS for ImageNet allows us to isolate the effect of posterior samples without noise from the base transitions.

SMC has three main parameters: the number of Monte Carlo (MC) samples $K$ used to estimate the reward-weighted posterior, the softmax temperature $\tau$, and (for ImageNet) the maximum guidance time $t_{\max}$ after which guidance is disabled. We resample at every outer step (including the final step) with ESS threshold 1.0, and share noise across particles. We choose to use a constant number of MC samples across time for SMC unlike guidance, but using a variable number is possible. We ablate at a fixed total NFE budget. NFE is computed as

$$
\text{NFE}_{\text{SMC}} = B \cdot \Big( \underbrace{(1 + \mathbb{1}_{\text{cfg}})TS}_{\text{base cost}} + \underbrace{K(T-1)M}_{\text{MC cost}} \Big)
$$

where $B$ is the number of particles used (batch size), $T$ is the number of outer denoising steps, $S$ is the number of inner refinement steps per outer step, and $K$ is the constant number of MC samples. The $TS$ term accounts for the forward passes we need for the base transitions (2 if using CFG). $K(T-1)M$ accounts for posterior samples at each non-terminal outer step, we use $M = 1/5$ for ImageNet1k similar to Section D.1.3 and $M = 1$ for CelebA-64. For CelebA-64, we a single inner step ($S = 1$) for the Posterior Diamond Map as the base drift. The batch size $B$ is chosen so that the total NFE satisfies the target budget.

**Prompt Alignment.** We evaluate SMC on prompt alignment for both datasets. For CelebA-64, we provide qualitative results optimizing for CLIPScore (Hessel et al., 2021) in Figure 10. For ImageNet, we optimize ImageReward (Xu et al., 2023), and ablate over $K \in 1, 2, 4, 8$, with $B$ chosen so total NFE matches the budget, comparing against a DPS baseline. We evaluated over 50 prompts, and evalute among all particles, measuring both the mean ImageReward value and best ImageReward value among the terminal particles Figure 11.

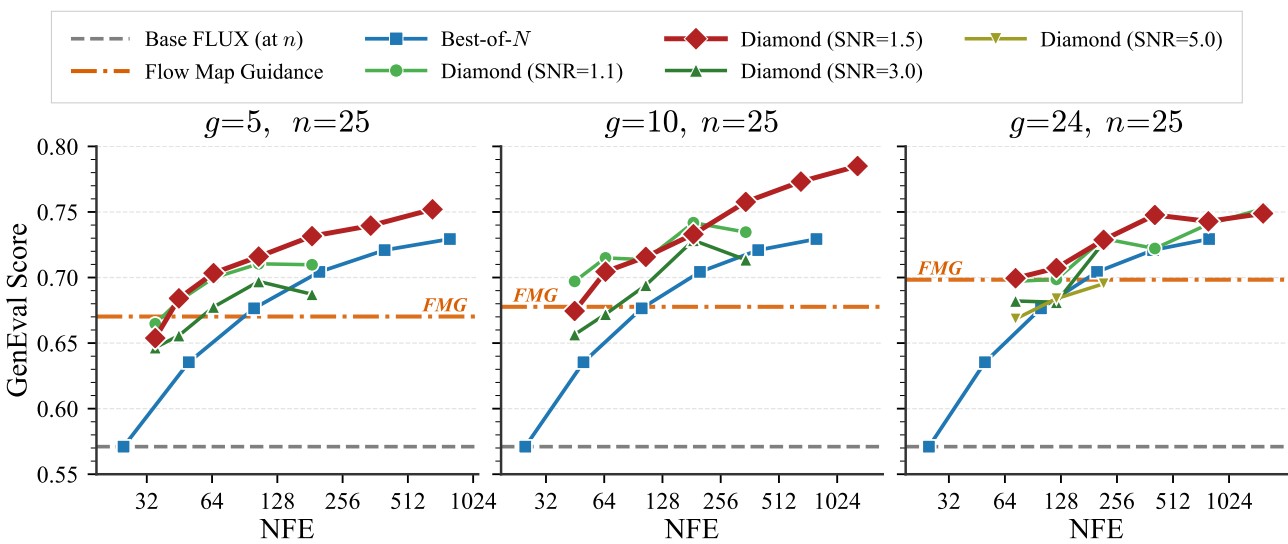

*Figure 16.* Ablation of hyperparameters on the FLUX Flow Map model ($n$=25). GenEval scores are reported across guidance steps $g$, perturbation noise level (SNR), and number of Monte Carlo samples MC. SNR=1.5 and $g$=10 yield the best trade-off.

*Table 6.* **Results on GenEval**. Comparison of existing inference-time reward-alignment methods with Weighted Diamond Maps. Weighted Diamond Maps outperform baseline methods, while being more efficient. Results with † taken from Eyring et al. (2025).

| Model | Time (s) ↓ | Mean ↑ | Single↑ | Two↑ | Counting↑ | Colors↑ | Position↑ | Attribution↑ |
|---|---|---|---|---|---|---|---|---|
| Flux-dev | 23.0 | 0.68 | 0.99 | 0.85 | 0.74 | 0.79 | 0.21 | 0.48 |
| SD3-Medium (Esser et al., 2024) | 4.4 | 0.70 | 1.00 | 0.90 | 0.72 | 0.87 | 0.31 | 0.66 |
| Qwen-Image (Wu et al., 2025) | 35.0 | 0.87 | 0.99 | 0.92 | 0.89 | 0.88 | 0.76 | 0.77 |
| SANA-Sprint 0.6B (Chen et al., 2025b) | 0.2 | 0.70 | 1.00 | 0.80 | 0.64 | 0.86 | 0.41 | 0.51 |
| + HyperNoise (Eyring et al., 2025)[†] | 0.3 | 0.75 | 1.00 | 0.88 | 0.71 | 0.85 | 0.51 | 0.55 |
| + Prompt Optimization (Mañas et al., 2024)[†] | 95.0 | 0.75 | 0.99 | 0.91 | 0.82 | 0.89 | 0.36 | 0.56 |
| + Best-of-N (Karthik et al., 2023)[†] | 15.0 | 0.79 | 0.99 | 0.92 | 0.72 | 0.91 | 0.53 | 0.65 |
| + ReNO (Eyring et al., 2024)[†] | 30.0 | 0.81 | 0.99 | 0.93 | 0.74 | 0.92 | 0.60 | **0.67** |
| + Weighted Diamond Maps | 10.0 | **0.83** | **1.00** | **0.95** | **0.84** | **0.94** | **0.61** | 0.65 |
| FLUX Flow Map (gabeguofanclub/flux-1-dev-flowmap-lsd)) | 11.0 | 0.57 | **1.00** | 0.66 | 0.65 | 0.66 | 0.15 | 0.31 |
| + Flow Map Guidance (Sabour et al., 2025b) | 46.0 | 0.74 | 0.99 | 0.89 | 0.80 | 0.91 | 0.21 | 0.62 |
| + Best-of-N (Karthik et al., 2023) | 703 | 0.75 | 0.99 | 0.91 | **0.85** | 0.87 | **0.30** | 0.61 |
| + Weighted Diamond Maps | 483 | **0.79** | **1.00** | **0.93** | 0.84 | **0.94** | 0.25 | **0.76** |

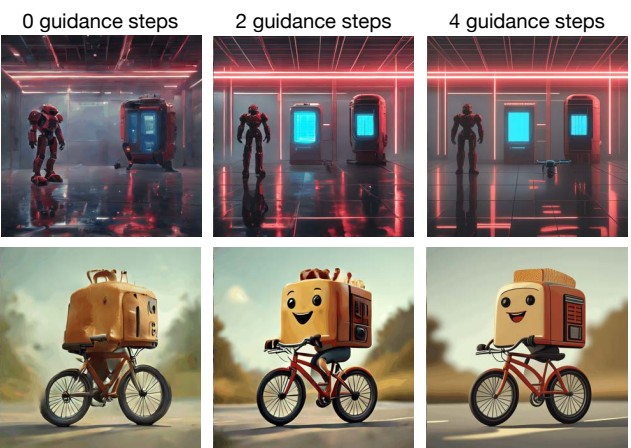

*Figure 15.* Illustration of guidance with Weighted Diamond Maps on SANA-Sprint. Trajectories of guidance (see Algorithm 3) are plotted after $N$ steps ($x_1$-prediction). Top: Prompt is "a matte red mech suit under dramatic key lights, a sleek electric blue drone hovering nearby, and a cylindrical maintenance pod with glowing panels". Note: The drone only appears after 4 guidance steps. Middle: "A toaster riding a bike". Guidance removed artifacts and increases adherence to prompt.

## D.2. Weighted Diamond Maps

**Implementation Details.** We simplify the importance weights from Theorem 5.1 to use only the reward, setting $w_k = \text{softmax}(r(z^1), \ldots, r(z^K))[k]$. The full weights additionally involve the recovery likelihood and a score correction, whose gradient norms both scale with the ambient dimension $d$ of the latent space and therefore dominate the reward gradient at high resolutions, making the composite weights difficult to balance numerically. All three gradient terms are however retained in the gradient estimate: we normalize each term individually to unit norm and rescale by a shared coefficient $\lambda$, placing them on a common scale regardless of dimension and controlling the overall guidance strength with a single hyperparameter. For SANA-Sprint, which directly predicts $x_1$, we recover the score and velocity via Tweedie's formula (24). We return the highest-reward sample among all flow-map rollouts; we apply the same selection rule to all baselines for a fair comparison. Full details are provided in Section D.2.

**Hyperparameters.** Weighted Diamond Maps has four main hyperparameters: the number of particles $K$, the number of guidance steps $N$, the renoising time $t' < t$, and the time interval over which guidance is applied. We set $b_t$ according to $b_t = \sigma_t^2 \frac{\dot{\alpha}_t}{\alpha_t} - \dot{\sigma}_t \sigma_t$.

For selecting the renoising time $t' < t$, we propose tying it to the signal-to-noise ratio. Letting $g(t) = \sigma_t^2 / \alpha_t^2$, we set $t' = g^{-1}(\lambda \cdot g(t))$, so that the SNR at $t'$ is a factor of $\lambda$ larger than at $t$. This reduces the renoising schedule to a single interpretable hyperparameter $\lambda$.

### D.2.1. NORMALIZATION OF GUIDANCE TERMS

In principle, Theorem 5.1 provides analytically derived weights for combining the score, likelihood, and reward gradient terms. However, in practice, these terms exhibit vastly different magnitudes: the score and likelihood gradients scale with the data dimensionality $d$, making them orders of magnitude larger than the reward gradient. Directly applying the analytical weights therefore leads to the reward signal being overwhelmed, resulting in negligible guidance effect.

To address this, we drop the analytical weights entirely and instead normalize each gradient term independently to unit norm before combining them. Concretely, let $g_{\text{score}}$, $g_{\text{likelihood}}$, and $g_{\text{reward}}$ denote the respective gradient contributions. We replace each with its unit-normalized counterpart:

$$\hat{g}_i = \frac{g_i}{\|g_i\|}, \quad i \in \{\text{score}, \text{likelihood}, \text{reward}\}. \tag{52}$$

This places all terms on equal footing regardless of their original magnitudes. The combined guidance signal is then scaled by a single shared hyperparameter $\lambda$:

$$\nabla_{x_t} V_t(x_t) \approx \lambda \sum_i \hat{g}_i. \tag{53}$$

We fix $\lambda = 20$ for all text-to-image experiments on SANA-Sprint and the Flux Flow Map. We observe that increasing $\lambda$ significantly beyond this value can cause the guidance term to dominate the base velocity, leading to visual artifacts in the generated images.

### D.2.2. FLUX FLOW MAP

**Model and inference.** We use the publicly available flow-map distillation of FLUX.1-dev (`gabeguofanclub/flux-1-dev-flowmap-lsd`). The base trajectory uses $n = 25$ ODE steps with step size $h = 1/n$. The model is parameterized as a flow map $X_{t,t'}^\theta(x_t)$ that accepts a source time $t$ and a target time $t'$. The base ODE step is performed by setting $t' = t + h$ using $X_{t, t+h}^\theta(x_t)$ to jump to the next step. To obtain the clean-endpoint prediction needed for reward evaluation, we set $t' = 1$, giving $\hat{x}_1 = X_{t,1}^\theta(x_t)$. This means the FLUX Flow Map needs two inference steps per Monte Carlo sample and guidance step.

**Hyperparameters.** We ablate the main hyperparameters for the FLUX Flow Map in Figure 16. Generally, Diamond Map scales well with more guidance steps, and a moderate SNR $\in [1.1, 1.5]$ seems to give the most robust result. We find the best compute-tradeoff for GenEval to be with SNR=1.5 and $g = 10, n = 25$.

### D.2.3. SANA-SPRINT ABLATIONS

**SANA-Sprint as a Flow Map.** SANA-Sprint is a consistency-distilled model designed for few-step generation that directly predicts the clean endpoint $\hat{x}_1$ from a noisy input $x_t$. Weighted Diamond Maps require access to both the endpoint prediction $\hat{x}_1(x_t, t)$ and a velocity estimate $v_t(x_t)$ at each inference step. Since SANA-Sprint already provides $\hat{x}_1$, we recover the velocity by applying Tweedie's formula to the flow map prediction at $t = 1$:

$$v_t(x_t) = \frac{\hat{x}_1(x_t, t) - x_t}{1 - t}. \tag{54}$$

This requires no retraining or architectural changes—we simply reinterpret the existing flow map output to extract both quantities needed by our framework. With the endpoint estimate and derived velocity in hand, we then apply Weighted Diamond Maps. Note, that this means for SANA-Sprint we only require $n + g * mc$ NFE, where $g$ is the guidance steps and $mc$ the amount of Monte Carlo samples.

*Table 7.* **Ablation of SNR and Particle Scaling**. Mean GenEval scores using 5 guidance steps $(0.05 \rightarrow 0.25)$. Total NFE $= 35$. The baseline configuration is highlighted.

| SNR ($\lambda$) | Particles ($P$) | | | | |
|---|---|---|---|---|---|
| | **1** | **2** | **4 (Default)** | **16** | **32** |
| 2.0 | 0.7775 | 0.8177 | 0.8202 | 0.8182 | 0.8207 |
| 5.0 | 0.7821 | 0.8118 | 0.8136 | 0.8089 | 0.8111 |
| 10.0 | 0.7896 | 0.8155 | 0.8067 | 0.8210 | 0.8223 |
| **20.0** | 0.8008 | 0.8095 | **0.8272** | 0.8239 | 0.8285 |

**SNR and Number of Particles.** We first ablate the SNR together with the number of particles while keeping guidance steps fixed to 5 in time-frame $[0.05, 0.25]$. As seen in Table 7, we find that while using a few particles $K > 2$ gives significant improvements, scaling the number of particles up, does not result in additional improvements.

*Table 8.* **Guidance Strategy Ablation**. Comparison of step counts and time horizons at SNR 20.0 and 4 Particles.

| Guidance Range | Steps ($N$) | NFE | Mean $\uparrow$ |
|---|---|---|---|
| **0.05 $\rightarrow$ 0.25 (Default)** | **5** | **30** | **0.8272** |
| 0.05 $\rightarrow$ 0.45 | 5 | 30 | 0.8163 |
| 0.05 $\rightarrow$ 0.25 | 10 | 60 | 0.8229 |
| 0.05 $\rightarrow$ 0.45 | 10 | 60 | 0.8284 |

**Number of guidance steps.** Next we ablate the number of guidance steps. As visualized in Figure 6c, we do see significant improvements when scaling up the number of guidance steps with diminishing returns after scaling to $N > 10$. Thus, we recommend keeping particles fixed to a range of [4, 8] while altering the number of guidance steps depending on available compute. For the time-frame, we find that $[0.05, 0.25]$ provides the better performance compared to longer time-frames with the same amount of steps, e.g. $[0.05, 0.45]$(Table 8).

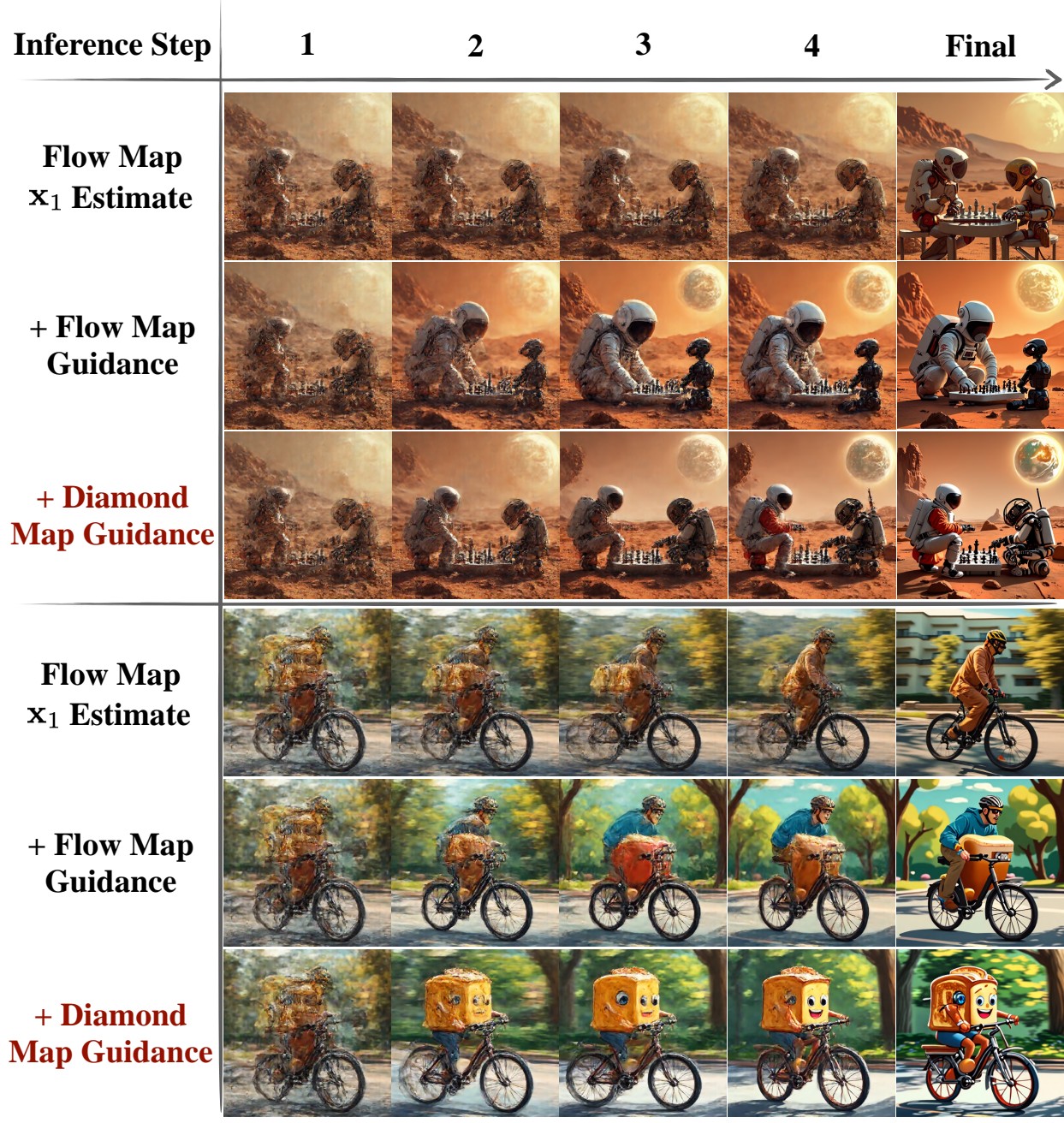

*Figure 17.* Progressive refinement across inference steps for FLUX Flow Map, comparing the flow map estimate $\hat{x}_1$, Flow Map Guidance, and Diamond Map Guidance. All rows share the same initial noise seed. Diamond Map Guidance yields sharper, more prompt-adherent results earlier in the trajectory.

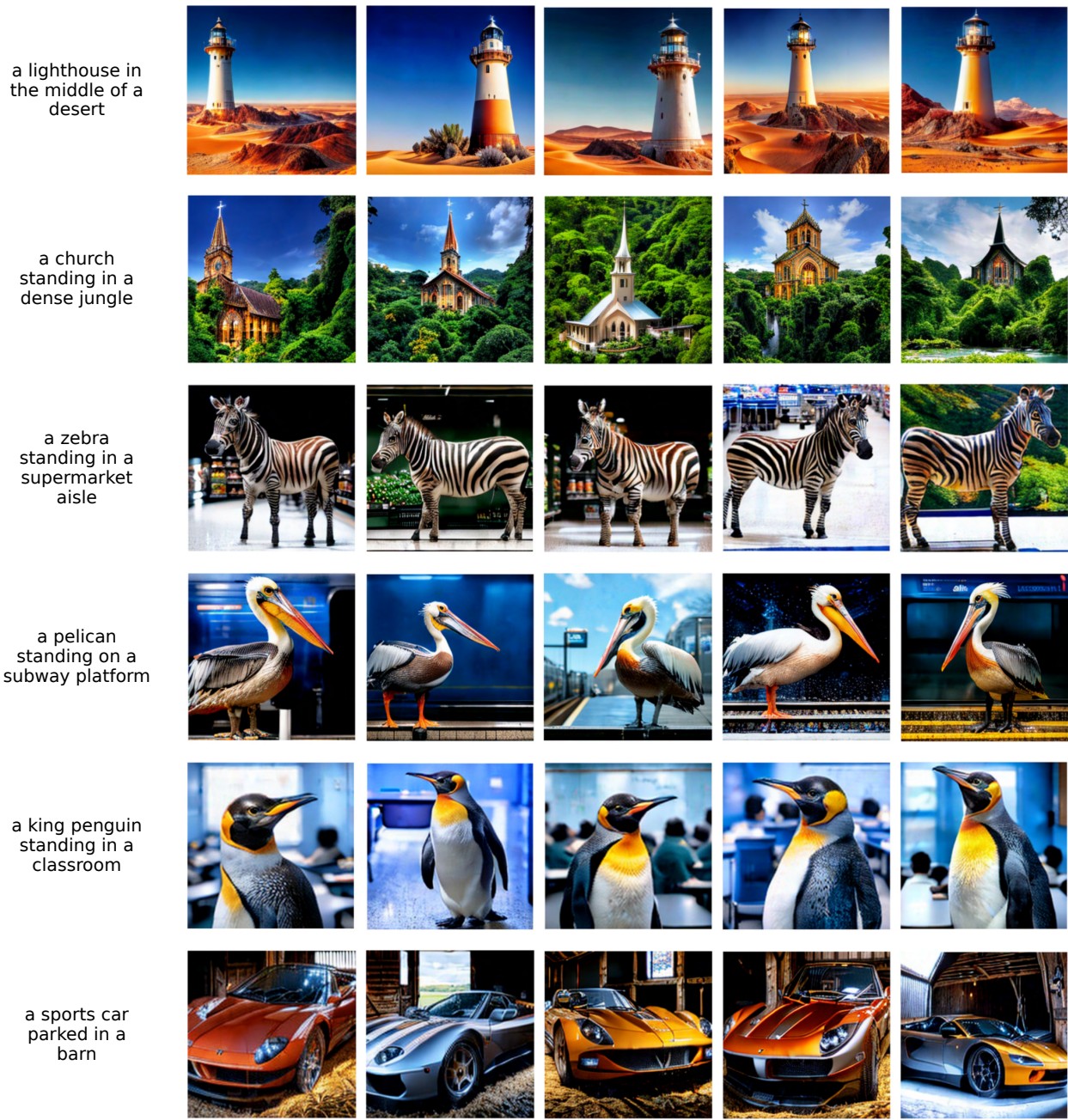

*Figure 19.* Making a class-conditional ImageNet model text-conditioned. Qualitative examples of ImageReward-reward alignment with guidance with Posterior Diamond Maps. Model trained on ImageNet1k (class-conditional). We extend class-conditioning to conditioning with simple prompts.

