# OpenReview forum: "Diamond Maps: Efficient Reward Alignment via Stochastic Flow Maps"
_ICML.cc/2026/Conference — ICML 2026 regular_

### Official Review · Reviewer_Y3Zw · 2026-03-07

**Soundness:** 4
**Presentation:** 4
**Significance:** 4
**Originality:** 3
**Overall Recommendation:** 5
**Confidence:** 4

**Summary:**

This paper addresses the task of inference-time reward alignment in diffusion / flow-based models, focusing the need of accurate and efficient posterior sampling $p _ {1|t}$. The authors propose posterior diamond maps, a one-step flow map model distilled from GLASS flows, and weighted diamond maps, a consistent value gradient estimator based on a normal flow map model. The paper provides empirical results of both methods on a variety of tasks including posterior sampling and inference-time reward alignment.

**Compliance With Llm Reviewing Policy:**

Affirmed.

**Final Justification:**

The authors' rebuttal has adequately addressed my concerns and I hope to see the paper accepted.

**Key Questions For Authors:**

1. The value gradient estimators in (15) and Prop. 5.1 require the differentiability of the reward function. What if the reward function is non-differentiable? Is it still possible to express the value gradient in terms of expectations over the one-step flow map, or derive a gradient-free estimator in the same spirit as Prop. 5.1? If so, how does the gradient-free estimators compare to gradient-based ones empirically in terms of bias and variance?
2. In the GLASS flow paper, the initial distribution of the inner flow matching model is $\mathcal{N}(\gamma x _ t,\sigma^2 I)$ for some $\gamma,\sigma$ that can be chosen arbitrarily. In this paper, the authors specifically focus on the case where the initial distribution is always standard Gaussian. Since we need to push forward the initial distribution to $p _ {1|t}(\cdot|x _ t)$, it is intuitively more efficient to choose the initial distribution to have some information about $x _ t$, so that the flow map is easier to learn. Have the authors considered other choices of the initial distribution, or is there a specific reason for choosing the standard Gaussian? (I understand one possible reason might be the double gradient $\nabla _ {x _ t}X _ {0,1}(\gamma x _ t + \sigma\epsilon;x _ t,t)$ when computing the value gradient estimator. Will it be less stable or more expensive to compute?)
3. In Prop. 4.3, the authors show that one can sample the *DDPM* transition kernel $p _ {t'|t}$ by the early stopped inner flow map. Since GLASS flow has a design choice $\rho$ that enables a series of transition kernels $p _ {t'|t}$ transporting $p _ t$ to $p _ {t'}$, with DDPM being a special case, are all these transition kernels realizable by the early stopped inner flow map?
4. Is there any deeper connection between the posterior diamond map and the normal flow map? For instance, is it possible to convert one to the other by some transformation? My question is motivated by the fact that the velocity field of the inner flow matching model can be expressed by the velocity field of the pretrained model using GLASS flow, and the flow maps correspond to the average velocity fields.
5. For the posterior diamond map, since it is a direct distillation of the pretrained flow matching model, do you require i.i.d. data samples from the dataset or can just use data sampled from the pretrained model? Please also comment on the learning error of the posterior diamond map, and how well it recovers the true posterior distribution (as the authors have mentioned in Sec. 4.2 that the flow maps have learning errors).
6. In SMC samplers, why do we choose $U _ {t,t'}(x _ t,x _ {t'})=V _ {t'}(x _ {t'})-V _ t(x _ t)$ instead of just $V _ {t'}(x _ {t'})$? Can the authors also comment on the ESS in (53) (since in the case where reward is sparse, the value function can be very peaky and concentrated on a small region)?
7. Minor comments: $\gamma _ i$ in Prop. 5.1: the gradient does not need transpose; the reference for FLUX (Labs, 2024) needs to be corrected.

**Limitations:**

See questions.

**Strengths And Weaknesses:**

**Strengths:**

- The two realizations of diamond maps are quite interesting and well-motivated for the task of inference-time reward alignment. The early stopped inner flow map sampling is a clever and intuitive way to sample transition $p _ {t'|t}$, and the weighted diamond map estimator is quite novel (although I don't have a good intuition for how it is derived).
- The paper is well-written and easy to follow, and is well positioned in the context of prior work such as GLASS flows and inference-time reward alignment methods. I have checked most of the proofs and derivations, and have not observed any major issues.
- Strong empirical results on a variety of tasks, which is expected given the benefits of flow maps over ODE sampling.

**Weaknesses:**

- The proposed two versions of diamond maps are quite different in terms of their design and motivation, and there is no strong connection between them. It would be nice to have a more unified perspective on these two methods. The choice involves a trade-off between the cost for distillation and the cost at inference time.
- The name "diamond maps" is not very intuitive to me and I don't see a strong connection to the method itself. A contemporary work arXiv:2601.14430 proposed similar flow-map models and called them "meta flow maps", which I find more descriptive.

---

> ### Author Rebuttal · Authors · 2026-03-31
>
> Thank you for your constructive feedback. We are pleased that Diamond Maps is considered “well-written and easy to follow” with “strong empirical results on a variety of tasks”.
>
> **Connection between Posterior Diamond Maps and Weighted Diamond Maps.**
>
> The unifying question our paper aims to address is: how can one perform efficient value function estimation using flow maps? This naturally leads to two complementary design points:
> 1. Weighted Diamond Maps ask whether an existing normal flow map can already be reused for efficient value estimation, without retraining a posterior-specific map.
> 2. Posterior Diamond Maps ask how one should redesign and distill a flow map so that posterior sampling and value estimation become as efficient as possible at inference time.
>
> Thus, the two methods address the same goal at different points of the training-vs-inference cost trade-off. We have revised the paper to present this unifying perspective explicitly. Please let us know whether there is a different way to address your concern.
>
>
> **Name of Diamond Maps.**
>
> *Diamond* Maps significantly reduce the computational cost of value function estimation compared to *Glass* Flows. A diamond is a more valuable material than glass — the name meataphorically captures the idea of refining Glass Flows into a more valuable (but harder to obtain) Diamond Map.
>
> Further, we note that while the concurrent “meta flow maps” work also explores the idea of training a flow map to estimate the value function, major theoretical and methodological innovations of our work - DDPM early stop sampling (section 4.2.) and the Weighted Diamond Map (section 5) - are not found in their work.
>
> **Additional experiments:**
>
> We have provided significantly more experiments, including on a more challenging ImageNet-256x256 task. To save space to answer your questions (we need to stay within the character limit), we refer to rebuttal for reviewer 8n3e. We would appreciate if these results would be taken into consideration for your final score.
>
>
> **Key Questions For Authors:**
>
> 1. **Differentiability:**
> Yes. Even when $r$ is non-differentiable, one can estimate $u_t^r(x)$ (and thereby $\nabla_{x_t}V(x_t)$) via a gradient-free identity (see Theorem 3.1. in [1]), where the expectation is over the posterior learnt via the Posterior Diamond Map. We have added this discussion.
> 1. **Choice of gamma and sigma:**
> In the posterior case ($t’=1$), GLASS fixes $\gamma=0$ since $\sigma_{t’}=0$. The remaining freedom is the scale parameter. We used the standard Gaussian for simplicity; varying it corresponds to rescaling the probability path and we observed no consistent benefit.
>
> 1. **GLASS distillation for general transition:**
> Not all GLASS transitions are realizable via early stopping. Prop. 4.3 is specific to the DDPM transition, which corresponds to the Markovian reverse of forward diffusion. More general transitions could be distilled but would require additional conditioning (e.g. $t’$, $\rho$). Our Sec. 4.2 avoids this, keeping the learning problem simpler.
>
> 1. **Convert flow map to posterior diamond map:**
> We do not believe such a transformation exists: while GLASS is linear at the velocity-field level, flow maps integrate those dynamics, inducing highly nonlinear dependence. The closest connection is Weighted Diamond Maps, which reuse a normal flow map for a *weighted* posterior estimator.
>
> 1. **Need of i.i.d data samples:**
> No. Teacher targets can be generated from the pretrained base model with the corresponding posterior construction, though this distills the pretrained model’s posterior rather than the true data posterior. Regarding learning error: Posterior Diamond Maps are subject to approximation error like all flow maps, but our results indicate this error is small enough to be effective. We will add a discussion clarifying this.
>
> 1. **SMC - choice of potential:**
> For exactness, potentials must sum to $r(x_1)$ ([2]). Our choice ensures this via $r(z_1)=\sum_{t}V_{t+h}(x_{t+h})-V_t(x_t)$, using $V_{0}(x_0)=0$. Using $V_{t’}(x_{t’})$ alone double-counts value from the proposal, causing overly aggressive reweighting. Regarding ESS in (53): it is good when the reward’s domain is well covered by the data distribution, but can be low when $r(x_1)$ concentrates outside the data support, at which point the problem becomes intractable. We have added a discussion of this limitation.
>
> We note that due to the character limit, we cannot address the questions in the desired detail. We are happy to answer any further  questions or provide more detail in the follow-up response.
>
> We appreciate the reviewer’s time and constructive comments. If the concerns are addressed, we would appreciate it if the reviewer would consider updating their score.
>
>
> [1] Feng et al., "On the guidance of flow matching," arXiv:2502.02150, 2025.
>
> [2] Singhal et al., "A general framework for inference-time scaling and steering of diffusion models," arXiv:2501.06848, 2025.

---

> > ### Author Rebuttal · Reviewer_Y3Zw · 2026-04-01
> >
> > I thank the authors for the detailed reply and the additional experimental results, which have fully resolved my concerns. This is a nice paper. I will maintain my positive rating and hope to see it accepted.
> >
> > Regarding the choice of $\gamma$ in GLASS flow, what I originally thought about (correct me if I was wrong) was that, even when $t'\ne1$, one has the flexibility to choose an arbitrary initial scale in the mean. Specifically, in the GLASS flow paper's Eqs. (16, 17), one can derive the conditional Gaussian distribution of $x_{t'}|x_t,z$ where there's a $\bar\gamma$, and in Eq. (18), when constructing the inner flow matching, we can actually use $p_s(\bar x_s|x_t,z)={\cal N}(\bar x_s;\bar\alpha_s z+{\color{red}\bar\gamma_s}x_t,\bar\sigma_s^2I)$ for some $\bar\gamma_s$ that satisfies $\bar\gamma_1=\bar\gamma$, and this includes the special case where $p_0(\bar x_0|x_t)={\cal N}(\bar x_0;0,\bar\sigma_0^2I)$. However, as the authors have reminded me that $\bar\gamma=0$ when $t'=1$, this is no longer an important issue here.

---

> > > ### Author Response · Authors · 2026-04-03
> > >
> > > We thank the reviewer for their response.
> > >
> > > **Re GLASS Flows:** GLASS Flows works for any "inner" Gaussian probability path $p_s(\bar x_s|x_t,z)={\cal N}(\bar x_s;\bar\alpha_s z+\bar\gamma_sx_t,\bar\sigma_s^2I)$. The only requirement is that $\bar\alpha_{0}=0$  (as one has to "forget" the data point at $s=0$ as it is unknown). In particular, one can vary:
> > > - $\bar\gamma_{s}$: Note that in the GLASS Flows paper, the authors have chosen to set $\bar\gamma_s=\bar\gamma$ to be constant in $s$. One can show that one make $\gamma_{s}$ also time-dependent (this would change the GLASS Flows velocity field, however).
> > > - $\bar\sigma_s$: Further, the scale parameter $\bar\sigma_s$ can also be chosen freely, including $\bar\sigma_{s=0}$.
> > > We agree that both hyperparameters could be explored in future work.
> > >
> > > We appreciate that we have "fully resolved" your concerns. If there is anything else that we can provide to increase your rating, we would be happy to provide more details or provide more experimental results.

---

### Official Review · Reviewer_8n3e · 2026-03-11

**Soundness:** 2
**Presentation:** 3
**Significance:** 3
**Originality:** 3
**Overall Recommendation:** 4
**Confidence:** 4

**Summary:**

This work tackles the problem of sampling from reward-tilted distributions of a pretrained flow matching model, whose analytical velocity field involves an intractable value function. To estimate the velocity field, this work suggests to use stochastic flow maps, called Diamond Maps, that give a consistent estimate of the value function. Two realisations of Diamond Maps are proposed: the first one is Posterior Diamond Maps which can be obtained by distilling GLASS Flows, and the second one is Weighted Diamond Maps that use deterministic ODE with renoising to introduce stochasticity. The Diamond Maps can naturally be used for SMC, providing an approximation for the optimal intermediate marginals (and thus expected to have lower variance). In the experiments, both Posterior and Weighted Diamond Maps are evaluated on a set of reward alignment tasks.

**Compliance With Llm Reviewing Policy:**

Affirmed.

**Final Justification:**

Although I believe the strengths of this work outweigh the weaknesses, one of my main concerns (W1) is not fully resolved.

Given the high dimensionality of the image space, I think that the number of particles used in their additional experiment is too small for proper benchmarking; there must be serious path degeneracy. I believe benchmarking SMC in a smaller image domain (like CIFAR10, or even synthetic targets) with more particles will be needed to validate the effectiveness of the proposed method.

I believe the evaluation metric should be improved (at least for the new SMC results provided in the rebuttal). The image reward does not measure what you want to achieve: sampling from the target posterior. Probably measuring FID score against a class subset of a dataset (e.g., CIFAR10) would be a better evaluation metric for assessing whether the samples match the target posterior distribution.

**Key Questions For Authors:**

1. It seems the proposed methods are not compatible with general probability paths, limiting the applicability. This should be discussed (or please correct me if I'm wrong).
2. In figure 6, various $b_t$ values are tested. How did you actually vary the $b_t$ when it is fully characterized by $\alpha_t, \sigma_t$?
3. Can Proposition 4.3. be used for Weighted Diamond Maps as well? If not, why?

*Note: Given that the questions are rather minor, I don't think my evaluation of this paper is affected by the answers to these questions. However, addressing the major points in Weaknesses can definitely change my evaluation.*

**Limitations:**

Although the paper includes an impact statement (which is written vaguely), it does not include a discussion of limitations.

**Strengths And Weaknesses:**

### Strengths

1. This paper is in general well-written and easy to follow, while some minor issues exist (see below).
2. The Posterior Diamond Maps are a natural extension of GLASS Flow and the Weighted Diamond Maps are based on a straightforward idea of renoising. Both ideas are simple, straightforward (which I appreciate), and also principled.

### Weaknesses
1. The experiments are not enough. Since the goal of this work is to match the reward-tilted distribution, it would be better to include some toy experiments that actually measure the quality of samples. Some quantitative evaluations seem to be needed for SMC integration, since currently it is only evaluated qualitatively (Figure 7).
2. No error (e.g., std or error bar) information is provided throughout the paper. I'm not entirely sure if this is a concern since I'm not an expert in *Image* generation. However, in scientific research, I think it is always better to include error information across multiple runs with different seeds.
3. From what I see in Table 3, the scalability of the Posterior Diamond Map is slightly questionable. It seems harder to scale up than the deterministic flow maps, possibly because it tries to solve a harder learning problem. The increased FID gap against the Flow Map in CelebA-64 compared to CIFAR10 seems to explain this.
4. These are minor points regarding the writing: a) I think it would be better to include a short discussion at the end of the paper, discussing limitations and/or future works. b) The notations are a bit messy. Especially, both $N$ and $K$ are used for the number of MC samples, see eq. (14) and (15) for example. In section 4, $X^{\theta}\_{s,r}$ and $X\_{s,r}$ are used a bit inconsistently. c) There are some typos: footnote 1 (equ. -> eq.), line 233 left ($u\_r$ -> $\bar{u}\_r$), line 942 (Theorem 4.3 -> Proposition 4.3), line 955 ($x^K\_1$ -> $x^N\_1$), line 1186 ('Figure 6. . Applying ...'), etc.

*I think the strengths slightly outweigh the weaknesses and am thus inclined to accept.*

---

> ### Author Rebuttal · Authors · 2026-03-31
>
> We would like to thank the reviewer for their constructive and positive feedback.
>
> **Scalability of Posterior Diamond Maps.**
>
> ```Posterior Diamond Map  [...] seems harder to scale up than the deterministic flow maps [...]```
>
> To address this, we trained a Posterior Diamond Map on ImageNet at 256x256 resolution, a 16x increase in image size over the 64x64 experiments in the paper. We repeat the posterior experiments (section 6.1):
>
> | posterior time $t$ | Posterior Diamond Map | GLASS Flows |
> |---|---|---|
> | 0.1 | 9.93 | 203.53 |
> | 0.2 | 8.86 | 114.19 |
> | 0.3 | 8.05 | 67.81 |
> | 0.5 | 4.68 | 14.20 |
>
> This confirms that posterior distillation scales to higher resolutions. A gap between the flow map and Posterior Diamond Map remains (one-step FID: 10.76 vs. 6.16 on ImageNet), but we believe this is not a fundamental limitation for two reasons:
> 1. Our experiments use the smaller DiT-B/2 model due to compute constraints; we expect this gap to narrow with larger models.
> 2. The Posterior Diamond Map estimates the gradient of the value function. The base flow model $u_t(x)$ is used for the velocity (Algorithm 1, line 10).
>
> That said, we agree that learning Posterior Diamond Maps is harder than learning deterministic flow maps. This is why we introduced Weighted Diamond Maps as an alternative that reduces learning complexity at the expense of more inference-time cost.
>
> **Additional experiments.**
>
> ```Some quantitative evaluations seem to be needed for SMC integration [...].```
>
> We provide quantitative SMC evaluation on a challenging prompt-alignment task using the ImageNet-256x256 model. Since ImageNet is class-conditional, we enable alignment with text prompts at inference time via SMC. As a baseline, we use Feynman-Kac steering (FKS) [1], an SMC method with biased value function approximation. We compare against SMC with value functions estimated via Posterior Diamond Maps, measuring prompt alignment with ImageReward. All rows are compute-matched: more Monte Carlo samples for value function estimation are compensated by fewer SMC particles. Posterior Diamond Maps outperform FKS, with the best results at 6 Monte Carlo samples.
>
> | Method | MC samples | ImageReward |
> |---|---|---|
> | Feynman-Kac steering | 1 | 0.6958 |
> | Ours | 2 | 0.7132 |
> | Ours| 6 | 0.7547 |
> | Ours | 8 | 0.7309 |
>
> We also evaluate on linear inverse problems (32x super-resolution with 0.05 Gaussian noise, bicubic downsampler) using Algorithm 1. We compare against FMTT [2], the state-of-the-art flow map guidance method, reporting LPIPS across varying compute budgets (NFEs). Posterior Diamond Maps outperform FMTT, especially at high compute budgets, because FMTT can only increase guidance steps, while our method can allocate NFEs toward more accurate estimators.
>
> | NFEs | FMTT [2] | Ours |
> |---|---|---|
> | 32 | 0.626 | 0.622 |
> | 64 | 0.599 | 0.598 |
> | 128 | 0.593 | 0.592 |
> | 256 | 0.597 | 0.588 |
> | 512 | 0.611 | 0.582 |
> | 1024 | 0.634 | 0.581 |
>
> **Error bars.**
>
> We note that GenEval [3] is a standard benchmark for text-to-image evaluation, where results are reported as the mean over 4 random seeds (which we followed in our experiments). This is the established protocol in prior work. To address this, we evaluated each seed individually for Table 1. Best-of-N achieves individual seed scores of 0.787, 0.808, 0.805, and 0.776 (mean 0.794 $\pm$ 0.013), while Weighted Diamond Maps achieves 0.835, 0.821, 0.826, and 0.830 (mean 0.828 $\pm$ 0.005). The improvement of Weighted Diamond Maps over Best-of-N is statistically significant relative to either method's standard deviation.
>
> **Key Questions For Authors:**
>
> 1. **General probability paths:** Our methods apply to general probability paths with one caveat: Posterior Diamond Maps could still be trained - however, not via distillation from Glass flows but by training from scratch (see section A.5.) because Glass flows rely on Gaussian probability paths.
>
> 2. **Choice of b_t in guidance:** The theoretically exact scalar term b_t is indeed known (see equation (11)). However, due to estimation error and high variance of the estimators for t<=0.3, it is common practice in guidance to tune b_t as a hyperparameter (see e.g. [1]). We have made this more explicit in our manuscript.
>
> 3. **Using Proposition 4.3. for Weighted Diamond Maps:** Indeed, proposition 4.3. can be used for Weighted Diamond Maps as well. However, the samples will be weighted (i.e. each sample is weighted). This could be integrated into an SMC pipeline (by updating the particle weights). We appreciate this suggestion. We will add a comment about this in our work.
>
> We appreciate the reviewer’s time and constructive comments. If the concerns are addressed, we would appreciate it if the reviewer would consider updating their score.
>
> [1] Singhal et al., "A general framework for inference-time scaling and steering of diffusion models."
>
> [2] Sabour et al., "Test-time scaling of diffusions with flow maps."
>
> [3] Ghosh et al., "GenEval"

---

> > ### Author Rebuttal · Reviewer_8n3e · 2026-04-03
> >
> > Thank you for your responses.
> >
> > > As a baseline, we use Feynman-Kac steering (FKS) [1]
> >
> > I don't really get the setup and the result. 1) Did you use gradient guidance for FKS? 2) How many particles did you use for FKS? 3) high image reward not necessarilly means better sampling
> >
> > > we agree that learning Posterior Diamond Maps is harder than learning deterministic flow maps
> >
> > >  Glass flows rely on Gaussian probability paths.
> >
> > Please discuss these points in the revised manuscript as well.
> >
> > I will keep my score at this point.

---

> > > ### Author Response · Authors · 2026-04-03
> > >
> > > We thank the reviewer for the thoughtful follow-up. We address each point below and have updated the manuscript accordingly.
> > >
> > > **Details on additional SMC experiments.**
> > >
> > > **1) Gradient guidance for FKS:** We do not use gradient guidance. We use the SMC algorithm outlined in Algorithm 2. This is a deliberate choice: our goal is to isolate the effect of improved value function estimation (via Posterior Diamond Maps with MC samples > 1) from orthogonal improvements like gradient guidance. We note that gradient guidance could be combined with our method and would likely improve results further. This is an advantage of our framework, not a limitation.
> > >
> > > **2) Number of particles for FKS:** We fix the total compute budget at $2000$ neural network evaluations (NFEs) across all rows and adjust the number of particles accordingly. As we increase the number of MC samples for value function estimation, the number of particles decreases. This leads to a range of number of particles of $6-12$. This compute-matched comparison is the fairest way to evaluate whether better value function estimates (our contribution) translate into better performance, and the results confirm they do.
> > >
> > > **3) ImageReward and sample quality:** We appreciate this important point. To clarify: we are not claiming that high ImageReward alone implies better image quality. ImageReward is used here specifically to measure *text-image alignment*. In this experimental setup, the goal is to steer a class-conditional ImageNet model toward alignment with text prompts at inference time that extend the class prompt - ImageReward directly measures success at this task. Importantly, since we do not use gradient guidance, we do not observe reward hacking artifacts (e.g., adversarial patterns that inflate reward without improving perceptual quality). We provide qualitative samples under the following anonymized link illustrating our experiments and the high image quality:
> > >
> > > https://drive.google.com/file/d/1PJ3x_Ab-OzNwRilVlG82hdg3HFUUqkjM/view?usp=sharing
> > >
> > > We also want to note that the goal of reward alignment is not to optimize image quality (usually measured by FID) but to adapt a pre-trained model to a new task — here, aligning class-conditional generations with specific text prompts. FID would not be the appropriate metric for this setting, as it measures distributional similarity to the original data rather than alignment with the target reward.
> > >
> > > **Gaussian probability paths/GLASS Flows and learning complexity of Posterior Diamond Maps.** We have added a discussion of these points to the revised manuscript. Specifically, we clarify that our framework applies to general probability paths — the restriction to Gaussian paths only arises when distilling Posterior Diamond Maps from GLASS Flows. We have also expanded the discussion of the learning complexity of Posterior Diamond Maps and how Weighted Diamond Maps allows us to remedy that.
> > >
> > > In summary, we believe the three experimental concerns have been addressed: (1) no gradient guidance is used, ensuring a controlled comparison; (2) all methods are compute-matched for fairness; and (3) ImageReward measures the intended objective, with qualitative results confirming perceptual quality. Combined with the new ImageNet-256x256 experiments from our previous response — which demonstrate scalability of both posterior distillation and SMC guidance — we hope the reviewer finds these concerns resolved.
> > >
> > > We would be grateful if the reviewer would consider raising their score in light of these clarifications, the significant theoretical innovations of our work, existing experiments on large-scale models, and the substantial new experiments.

---

### Official Review · Reviewer_cyFw · 2026-03-14

**Soundness:** 4
**Presentation:** 3
**Significance:** 4
**Originality:** 3
**Overall Recommendation:** 5
**Confidence:** 4

**Summary:**

The authors present Diamond Maps, a method that enables reward calculation for flow models. The authors begin by constructing a stochastic flow map. This stochastic flow map enables efficient evaluation of the reward function and its gradient for a partially denoised sample xt. Then the authors continue to introduce weighted diamond maps, in which the authors use a re-noising techniqe paired with an addition to the loss term to sample from the posterior p1(.|xt) efficiently. Both techniques allow for the evaluation of the value function Vrt(xt). The authors evaluate their method on CelebA and CIFAR10, and use their method on a prompt alignment challenge.

**Compliance With Llm Reviewing Policy:**

Affirmed.

**Final Justification:**

My initial review of this paper was positive, and the rebuttal the authors provided answered my questions and raised no new concerns. I therefore maintained my already positive score.

**Key Questions For Authors:**

See strenghts and weaknesses

**Limitations:**

The authors have not discussed limitations or societal impact

**Strengths And Weaknesses:**

I thoroughly enjoyed reading this paper, I think the method is intuitive and the authors do a good job explaining a fairly complex subject. It could also prove an incredibly useful method for practitioners. The paper not only proposes a distillation method, but also a sampling based approach, giving users the option whether they want to spend compute budget at training time or inference time. Here are some issues I found with the paper:
- I find the overbar notation quite confusing, e.g. bar(x0) on the left in eqn 13. Could the authors elaborate on what it means? I can't figure out the pattern, it comes across as arbitrary. It would help the paper if they mentioned it there too.
- Labelling an intermediate time r with a reward function floating around is also a confusing choice.
- It's not clear to me what the point of section 4.2 is. I think the strength of the method lies in evaluation the value function, and I don't understand why and how it is useful to sample from the distribution itself. Would one not just use Eqn 11 together with 15 to create samples? I'm curious to know how the authors think this fits into the value function picture, or whether it is just a separate approach?
- As far as I can tell, an experiment on distilled the distilled value function estimation is missing, in my opinion that would be a lot more useful to the message of the paper than the evaluation of the quality of the samples itself, e.g. similar results to table 1, but with a distilled model. In my opinion, this distilled value function estimation could be very powerful for applications and it would be useful to see it evaluated in the paper, much more useful than few-step generation, for which there already exist plenty of powerful methods.

(not an issue, but there is a typo on line 383 "it not fully determined" )

---

> ### Author Rebuttal · Authors · 2026-03-31
>
> We would like to thank the reviewer for their constructive and positive feedback. We are pleased to see that the reviewer thinks that Diamond Maps “could prove an incredibly useful method for practitioners” and the paper does “a good job explaining a fairly complex subject”.
>
> **Significance of section 4.2**
>
> ```It's not clear to me what the point of section 4.2 is. [...] I'm curious to know how the authors think this fits into the value function picture, or whether it is just a separate approach?```
>
> Thank you for raising this point. The key insight of section 4.2 is that Posterior Diamond Maps accelerate any reward alignment method that relies on DDPM transitions [1,2,3,4,5] — beyond just value function estimation. Concretely, we prove that the Posterior Diamond Map contains a flow map that samples transitions from the DDPM/time-reversal SDE, even though it was only trained to sample from the posterior. This is relevant because value function estimation is only one part of the compute cost in many alignment methods; the other bottleneck is sampling DDPM transitions. For example:
> - In Monte-Carlo tree search [1,2], transitions are needed to *sample nodes in the tree* (step 1), while value functions are used to *evaluate* them (step 2). Section 4.2 accelerates step 1.
> - In Sequential Monte Carlo [3,4,5], value function is used to construct the SMC potential (step 1), while DDPM transitions serve as the proposal distribution (step 2). Section 4.2 accelerates step 2.
>
> We have updated our manuscript to better highlight the role and significance of section 4.2.
>
> **Additional experiments with Posterior Diamond Maps:**
>
> ```As far as I can tell, an experiment on distilled the distilled value function estimation is missing [...]```
>
> We provide quantitative SMC evaluation on a challenging prompt-alignment task using the ImageNet-256x256 model (we trained a new Posterior Diamond Map to tackle more challenging benchmarks, see response to reviewer 8n3e). Since ImageNet is class-conditional, we enable alignment with text prompts at inference time via SMC. As a baseline, we use Feynman-Kac steering (FKS) [4], an SMC method with biased value function approximation. We compare against SMC with value functions estimated via Posterior Diamond Maps, measuring prompt alignment with ImageReward. All rows are compute-matched: more Monte Carlo samples for value function estimation are compensated by fewer SMC particles. Posterior Diamond Maps outperform FKS, with the best results at 6 Monte Carlo samples.
>
> | Method | MC samples | ImageReward |
> |---|---|---|
> | Feynman-Kac steering | 1 | 0.6958 |
> | Ours | 2 | 0.7132 |
> | Ours| 6 | 0.7547 |
> | Ours | 8 | 0.7309 |
>
> We also evaluate on linear inverse problems (32x super-resolution with 0.05 Gaussian noise, bicubic downsampler) using Algorithm 1. We compare against FMTT [7], the state-of-the-art flow map guidance method, reporting LPIPS across varying compute budgets (NFEs). Posterior Diamond Maps outperform FMTT, especially at high compute budgets, because FMTT can only increase guidance steps, while our method can allocate NFEs toward more accurate estimators.
>
> | NFEs | FMTT [7] | Ours |
> |---|---|---|
> | 32 | 0.626 | 0.622 |
> | 64 | 0.599 | 0.598 |
> | 128 | 0.593 | 0.592 |
> | 256 | 0.597 | 0.588 |
> | 512 | 0.611 | 0.582 |
> | 1024 | 0.634 | 0.581 |
>
>
> **Notation:**
> - *Overbar notation:* The overbar distinguishes the inner state $\bar{x}\_s$ (evolving within the posterior) from the outer state $x\_t$ (fixed conditioning point), following the convention of GLASS Flows [6]. This translates to the flow map $\bar{X}\_{s,s'}(\bar{x}\_0|x\_t,t)$. We agree with the reviewer that this might be challenging notation and have added clearer descriptions in the manuscript.
> - *Confusion of intermediate time r with a reward function:* We agree with the reviewer that this might be confusing to some readers and updated our manuscript to remove any indices for time $r$ and reserve “r” for reward functions.
> - *Typo:* We thank the reviewer for catching the typo on line 383 ("it not fully determined") and have corrected it.
>
> **Limitations.** We have added a discussion of limitations and societal impact to the revised manuscript.
>
> We appreciate the reviewer’s time and constructive comments, which have helped strengthen the manuscript. If the concerns are addressed, we would appreciate it if the reviewer would consider updating their score.
>
> [1] Li et al. Dynamic search for inference-time alignment in diffusion models.
>
> [2] Zhang et al. Inference-time scaling of diffusion models through classical search, 2025.
>
> [3] Wu et al. Practical and asymptotically exact conditional sampling in diffusion models.
>
> [4] Singhal et al. A general framework for inference-time scaling and steering of diffusion models.
>
> [5] Skreta et al. Feynman-kac correctors in diffusion: Annealing, guidance, and product of experts.
>
> [6] Holderrieth et al. GLASS Flows.
>
> [7] Sabour et al., "Test-time scaling of diffusions with flow maps."

---

> > ### Author Rebuttal · Reviewer_cyFw · 2026-04-04
> >
> > I thank the authors for their thoughtful response, they have addressed my questions adequately, I will maintain my score.

---

### Decision · Program_Chairs · 2026-04-30

**Decision:**

Accept (regular)

**Comment:**

This paper studies the problem of sampling posteriors under flow model priors. Two schemes are proposed for approximating the value function (and, more importantly, its gradient, which gives the guidance term for sampling the posterior) using estimators derived from iterating amortised integration of the flow and renoising.

The reviewers found the idea intuitive, well-presented, and principled. No important weaknesses remain unaddressed or questions unanswered following the rebuttal and discussion, except the request from 8n3e regarding better evaluation of SMC (with which I agree -- posterior fit is not currently well demonstrated).

The authors should take all reviewers' comments into account in the final version. To add, for accessibility, please do not use colour alone (especially red/green, like in Figure 3) to make essential distinctions in figures.